# Geodesic Graph Neural Network for Efficient Graph Representation Learning

**Lecheng Kong**
Washington University in St. Louis
`jerry.kong@wustl.edu`

**Yixin Chen**
Washington University in St. Louis
`ychen25@wustl.edu`

**Muhan Zhang**
Peking University
`muhan@pku.edu.cn`

## Abstract

Graph Neural Networks (GNNs) have recently been applied to graph learning tasks and achieved state-of-the-art (SOTA) results. However, many competitive methods run GNNs multiple times with subgraph extraction and customized labeling to capture information that is hard for normal GNNs to learn. Such operations are time-consuming and do not scale to large graphs. In this paper, we propose an efficient GNN framework called Geodesic GNN (GDGNN) that requires only one GNN run and injects conditional relationships between nodes into the model without labeling. This strategy effectively reduces the runtime of subgraph methods. Specifically, we view the shortest paths between two nodes as the spatial graph context of the neighborhood around them. The GNN embeddings of nodes on the shortest paths are used to generate geodesic representations. Conditioned on the geodesic representations, GDGNN can generate node, link, and graph representations that carry much richer structural information than plain GNNs. We theoretically prove that GDGNN is more powerful than plain GNNs. We present experimental results to show that GDGNN achieves highly competitive performance with SOTA GNN models on various graph learning tasks while taking significantly less time.

## 1 Introduction

Graph Neural Network (GNN) is a type of neural network that learns from relational data. With the emergence of large-scale network data, it can be applied to solve many real-world problems, including recommender systems [44], protein structure modeling [15], and knowledge graph completion [2]. The growing and versatile nature of graph data pose great challenges to GNN algorithms both in their performance and their efficiency.

GNNs use message passing to propagate features between connected nodes in the graph, and the nodes aggregate their received messages to generate representations that encode the graph structure and feature information around them. These representations can be combined to form multi-node structural representations. Because GNNs are efficient and have great generalizability, they are widely employed in node-level, edge-level, and graph-level tasks. A part of the power of GNNs comes from the fact that they resemble the process of the 1-dimensional Weisfeiler-Lehman (1-WL) algorithm [22]. The algorithm encodes subtrees rooted from each node through an iterative node coloring process, and if two nodes have the same color, they have the same rooted subtree and should have very similar surrounding graph structures. However, as pointed out by Xu *et al.* [45], GNN's expressive power is also upper-bounded by the 1-WL test. Specifically, GNN is not able to differentiate nodes that have exactly the same subtrees but have different substructures. For example, consider the graph

in Figure 1 with two connected components, because all nodes have the same number of degrees, they will have exactly the same rooted subtrees. Nevertheless, the nodes in the left component are clearly different from the nodes in the right component, because the left component is a 3-cycle and the right one is a 4-cycle. Such cases can not be discriminated by GNNs or the 1-WL test. We refer to this type of GNN as plain GNN or basic GNN.

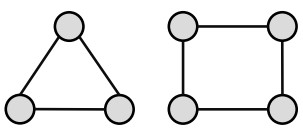

Figure 1: An example where plain GNN fails to distinguish nodes in the graph.

On the other hand, even when a carefully-designed GNN can differentiate all node substructures in a graph, Zhang *et al.* [54] show that the learned node representation still cannot effectively perform structural learning tasks involving multiple nodes, including link predictions and subgraph predictions, because the nodes are not able to learn the relative structural information. We will discuss this in more detail in Section 3.1.

These limitations motivate recent research that pushes the limit of GNN's expressiveness. One branch is to use higher-order GNNs that mimic higher-dimensional Weisfeiler-Lehman tests [23, 26]. They extend node-wise message passing to node-tuple-wise and obtain more expressive representations. Another branch is to use labeling methods. By extracting subgraphs around the nodes of interest, they design a set of subgraph-customized labels as additional node features. The labels can be used to distinguish between the nodes of interest and other nodes. The resulting node representations can encode substructures that plain GNNs cannot, like cycles.

However, the major drawback of these methods is that compared to plain GNNs, their efficiency is significantly compromised. Higher-order GNNs update embeddings over tuples and incur at least cubic complexity [52, 23]. Labeling tricks usually require subgraph extraction on a large graph and running GNNs on mini-batches of subgraphs. Because the subgraphs overlap with each other, the aggregated size of the subgraphs is possibly hundreds of times the large graph. Nowadays GNNs are massive, the increased graph computation cost makes labeling methods inapplicable to real-world problems with millions of nodes.

We observe that a great portion of the labeling methods relies on geodesics, namely the shortest paths, between nodes. For node representation, Distance-Encoding (DE) [25] labels the target nodes' surrounding nodes with their shortest path distance to the target nodes and achieves higher than 1-WL test expressiveness. Current state-of-the-art methods for link prediction, such as SEAL [50], leverage the labeling tricks in order to capture the shortest path distance along with the number of shortest paths between the two nodes of the link. Inspired by this, we propose our graph representation learning framework, Geodesic GNN (GDGNN), where we run GNN **only once** and inject higher expressiveness into the model using shortest paths information by a geodesic pooling layer. Our framework can obtain higher than 1-WL power without multiple runs of plain GNNs, which largely improves the efficiency of more expressive methods.

Specifically, our GDGNN model learns a function that maps the learning target to a geodesic-augmented target representation (the target can be a node, edge, graph, etc). The model consists of two units, a GNN unit, and a geodesic pooling unit. The two units are connected to perform end-to-end training. We first apply the GNN to the graph to obtain embeddings of all nodes. Then, for each target, we find the task-specific geodesics and extract the corresponding GNN representations of nodes on the geodesics. The task-specific geodesics have three levels. For node-level tasks, we extract the geodesics between the target node and all of its neighbors. For edge-level tasks, we extract the geodesics between the two target nodes of the edge. For graph-level tasks, we get the geodesic-augmented node representation for each node as in node-level tasks and readout all node representations as the graph representation. Finally, using the geodesic pooling unit, we combine the geodesic representation and target representation to form the final geodesic augmented target representation. Figure 2 summarizes the GDGNN framework.

All task-specific geodesics are formed by one or multiple **pair-wise** geodesics between two nodes. We propose two methods to extract the pair-wise geodesic information, horizontal and vertical. The horizontal geodesic is obtained by directly extracting one shortest path between the two nodes. The vertical geodesic is obtained by extracting all direct neighbors of the two nodes that are on any of their shortest paths. They focus on depth and breadth respectively. Horizontal geodesic is good at capturing long-distance information not covered by the GNN. Vertical geodesic is *provably more powerful* than plain GNN. Moreover, by incorporating the information of the subgraph induced by

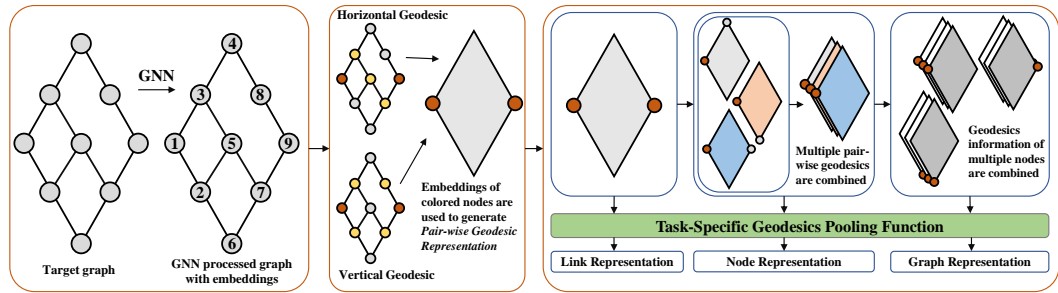

Figure 2: The overall pipeline of GDGNN. GNN is applied once to generate node embeddings. We then use horizontal or vertical geodesic to extract pair-wise geodesic representation. One or more such representations are collected to generate task-specific representation.

the vertical geodesic nodes, we can use GDGNN to distinguish some *distance-regular graphs*, a type of graph that many current most powerful GNNs cannot distinguish.

During inference, the GNN unit is applied only once to the graph, after which we feed nodes and geodesic information to the efficient geodesic pooling unit. This limits the number of GNN runs from the query number (or the number of nodes for graph-level tasks) to a constant of one and remarkably reduces the computation cost. *Compared to labeling methods, GDGNN only requires one GNN run and is much more efficient. Compared to plain GNN methods, GDGNN uses geodesic information and thus is more powerful.* We conducted experiments on node-level, edge-level, and graph-level tasks across datasets with different scales and show that GDGNN is able to significantly improve the performance of basic GNNs and achieves very competitive results compared to state-of-the-art methods. We also demonstrate GDGNN's runtime superiority over other more-expressive GNNs.

## 2 Preliminaries

A graph can be denoted by $\mathcal{G} = (\mathcal{V}, \mathcal{E}, \mathcal{R})$, where $\mathcal{V} = \{v_1, ..., v_n\}$ is the node set, $\mathcal{E} \subseteq \{(v_i, r, v_j) | v_i, v_j \in \mathcal{V}, r \in \mathcal{R}\}$ is the edge set and $\mathcal{R}$ is the set of edge types in the graph. When $|\mathcal{R}| = 1$, the graph can be referred to as a homogeneous graph. When $|\mathcal{R}| > 1$, the graph can be referred to as a heterogeneous graph. Knowledge graphs are usually heterogeneous graphs. The nodes can be associated with features $\mathcal{X} = \{\boldsymbol{x}_v | \forall v \in \mathcal{V}\}$. For simplicity, we use homogeneous graphs to illustrate our ideas and methods.

We follow the definition of message passing GNNs in [45] with AGGREGATE and COMBINE functions. Such GNNs iteratively update the node embeddings by receiving and combining the neighbor nodes' embeddings. The $k$-th layer of a GNN can be expressed as:

$$\boldsymbol{m}_u^{(k)} = \text{AGGREGATE}^{(K)}(\boldsymbol{h}_u^{(k-1)}), \quad \boldsymbol{h}_v^{(k)} = \text{COMBINE}(\{\boldsymbol{m}_u^{(k)}, u \in \mathcal{N}(v)\}, \boldsymbol{h}_v^{(k-1)}) \quad (1)$$

where $\boldsymbol{h}_v^{(k)}$ is the node representation after $k$ iterations, $\boldsymbol{h}_v^{(0)} = \boldsymbol{x}_v$, $\mathcal{N}(v)$ is the set of direct neighbors of $v$, $\boldsymbol{m}_u$ is the message embedding. Different GNNs vary mainly by the design choice of the AGGREGATE and COMBINE functions. In this paper, we use $\boldsymbol{h}_v$ to represent the final node representation of $v$ from the plain GNN. In this paper, we use bold letters to represent vectors, capital letters to represent sets, and $\mathcal{N}^k(v)$ to refer to nodes within the k-hop neighborhood of $v$.

## 3 Geodesic Graph Neural Network

Geodesic is conventionally defined as the shortest path between two nodes and indeed in homogeneous graphs, it encodes only the distance. However, we found that when combined with GNNs and node representations, geodesics express information much richer than the shortest path distance alone. In this section, we first discuss the intuition behind GDGNN and its effectiveness. Then, we describe the GDGNN framework and how its GNN unit and geodesic pooling unit are combined to make predictions. Finally, we provide theoretical results of the better expressiveness of GDGNN.

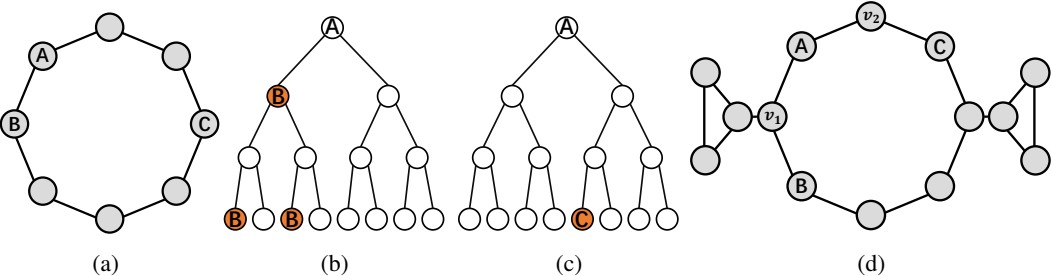

Figure 3: (a) A circular graph where plain GNN cannot distinguish link AB and AC. (b) and (c) are the GNN computation graphs of A if B and C, respectively, are labeled. (d) is an example where geodesic representation can distinguish links AB and AC, but distance fails to do so.

## 3.1 Intuition and Overview

The key weakness of directly using plain GNNs is that the embeddings are generated without conditional information. For node-level tasks, as pointed out in [49], a plain GNN is unaware of the position of the target node in the rooted subtree, and hence the target node is oblivious of the conditional information about itself. For edge-level tasks, Zhang *et al.* [54] shows that despite a node-most-expressive-GNN, without node labeling, the embedding of one node in the edge is not able to preserve the conditional information of the other node. Concretely, take link prediction as an example, in Figure 3a, we have two target links AB and AC. Since AB is already connected by a link and AC is not, they should be represented differently. On the other hand, All nodes in the graph are symmetric and should be assigned the same embeddings using any GNN. Hence, if we simply concatenate the node representations of AB and AC, they will have the same link representations, and cannot be differentiated by any downstream learner. To overcome this problem, previous methods rely on labeling tricks to augment the computation graphs with conditional information. When node B and C are labeled respectively in two separate subgraphs, the resulting computation graphs of A are different as shown in Figure 3b and 3c.

Notice that, for the labeling methods to take effect, subgraph extraction is necessary. However, relabeling the graph and applying a GNN to every target to predict is extremely costly. Hence, we wonder whether we can also incorporate conditional information between nodes without customized labeling. Observe that in the example, the computation graph captures the number of layers before node A first receives a message from a labeled node, which is essentially the shortest path distance $d$ between A and the other node. This implies that even when we **only** have node representations from a plain GNN, by adding the distance information (which is 1 and 3 for AB and AC respectively) independent of the computation graph, we can still distinguish the links.

While the previous example demonstrates the effectiveness of a simple shortest path distance metric, we notice that distance alone is not always sufficient. For example, in Figure 3d, node B and node C are symmetric and hence should be assigned the same node embedding when the graph is not labeled. However, links AB and AC are apparently different as $v_1$ connecting AB connects to another group of nodes whereas $v_2$ connecting AC does not connect to any other node. If we only use the shortest path distance between AB and AC to assist the prediction, we still cannot differentiate them. However, if we inject the **entire geodesic** expressed by the corresponding **node embeddings along the shortest path** to the model, we are able to tell AB and AC apart due to $v_1$ and $v_2$'s different embeddings.

Consider a naive GNN that outputs all node embeddings as one. Taking the shortest path can be seen as extracting a one-valued vector with a length of the shortest distance between the two nodes. In the neural setting, a general GNN can model the naive GNN and the nodes on the same shortest path can have different representations besides one by encoding their own neighborhood. The extension of the shortest path to the neural setting grants stronger expressive power to the geodesics. The geodesic generation and the GNN are *decoupled*, because the selection of geodesic nodes is independent of the node representations, while the node representations in subgraph GNNs are tied to a subgraph. Dwelling on this decoupling and conditioning philosophy, GDGNN can run the GNN **only once** to generate generic node embeddings and then use geodesic information to inject conditional information into the model in a much **more efficient** way.

## 3.2 Pair-wise Geodesic Representation

GDGNN has different ways to leverage geodesic information for different levels of tasks, but they all have the same basic component, pair-wise geodesic representation. It consists of structural information of nodes on the shortest paths of the pair. Directly extracting all such nodes is time-consuming and sometimes not effective, hence we present two primary ways to learn the geodesic representation, *horizontal* and *vertical*.

### 3.2.1 Horizontal Geodesic Representation

Horizontal Geodesic Representation is designed to directly capture the distinctiveness as described in the previous example shown in Figure 3d. It first finds the shortest path between two nodes as the geodesic. If there are multiple paths between them, we randomly choose one. The nodes on the shortest path is denoted by $P_{(u,v)}^{(H)} = \{u, n_1, n_2, ..., n_{d-1}, v\}$ with a length of $d$. Using $P_{(u,v)}^{(H)}$ we can construct the geodesic representation $\boldsymbol{g}_{(u,v)}$ by a geodesic pooling layer.

$$\boldsymbol{g}_{(u,v)}^{(H)} = R_{gd}^{(H)}(\{\boldsymbol{h}_w, \forall w \in P_{(u,v)}^{(H)}\}) \tag{2}$$

To prevent extremely long geodesics which may cause overfitting, we limit the maximum length to a manageable constant $d_{max}$. Any pair with a distance larger than $d_{max}$ will have an infinite distance and a zero geodesic representation.

Because of the high computation cost of GNNs, the number of GNN layers is usually very small. Using horizontal geodesics, we are able to connect nodes that are *far away* and do not coexist in a GNN's limited-height computation graph.

### 3.2.2 Vertical Geodesics Representation

Horizontal geodesics seems like an intuitive choice to directly connect the pair of nodes by a sequence of adjacent nodes. However, a natural question to ask is whether a single path is enough to represent the conditional information between the pair. An example where horizontal geodesic falls short can be found in Appendix G. Realizing this limitation, we propose the vertical geodesic representation that includes all geodesic information in an efficient way.

For a pair of nodes $u$ and $v$, vertical geodesic extracts the **direct neighbors** of them that are on **any** of their shortest paths. Specifically, let the distance between $u$ and $v$ be $d(u,v)$, we have the nodes on their shortest paths to be $W_{(u,v)} = \{w | d(u,w) + d(w,v) = d(u,v), \forall w \in \mathcal{V}\}$. The geodesics can be represented as,

$$P_{(u,v)}^{(V)} = P_{(u,v),u}^{(V)} \cup P_{(u,v),v}^{(V)}, \quad \text{where } P_{(u,v),i}^{(V)} := W \cap \mathcal{N}(i), i \in \{u, v\}. \tag{3}$$

Here $P_{(u,v),i}^{(V)}$ represents the nodes in the geodesic that are also the direct neighbors of $i$. For some tasks, we only consider one $P_{(u,v),i}^{(V)}$ as the geodesics instead of both. We use a pooling layer to generate vertical geodesic

$$\boldsymbol{g}_{(u,v)}^{(V)} = R_{gd}^{(V)}(\{\boldsymbol{h}_w, \forall w \in P_{(u,v)}^{(V)}\}). \tag{4}$$

While the vertical geodesic does not explicitly encode a long shortest path, it can still capture short shortest path information due to pooling the embeddings of the nodes in it. Compared with the horizontal geodesic that takes one random shortest path, the vertical geodesic focuses more on breadth by incorporating all direct neighbors on any shortest paths.

Unlike nodes in horizontal geodesic, nodes in vertical geodesic induce a more complex subgraph. Consider only $P_{(u,v),v}^{(V)}$ (we will use $P_v$ here for simplicity) and the subgraph $\mathcal{G}_v$ induced by nodes in $P_v$. We can have a pooling function that encodes the graph structure of $\mathcal{G}_v$,

$$\boldsymbol{g}_v^{(V)} = R_{gd}^{(G)}(\{\boldsymbol{h}_w, \forall w \in P_v\}, \mathcal{G}_v). \tag{5}$$

The pooling layer can be a small plain GNN itself, and the initial embedding of $\mathcal{G}_v$ can take the pooled node embeddings from the main GNN. However, to keep our method efficient, we only add

the degree of nodes in the subgraph to the node embeddings. And the vertical geodesic representation becomes,

$$\boldsymbol{g}_v^{(V)} = R_{gd}^{(V)}(\{\boldsymbol{h}_w \oplus deg_{\mathcal{G}_v}(w), \forall w \in P_v\}), \tag{6}$$

where $\oplus$ means concatenation. Note that a node's degree in the subgraph is different from that in the main graph. This can be seen as a 1-layer GNN applied onto $\mathcal{G}_v$ with identical initial node embeddings. Even with this simple pooling design, the vertical geodesic is able to help distinguish some distance regular graphs, which we cover in section 3.4.

Because vertical geodesics do not encode distance directly, we concatenate distance to vertical geodesics as an additional feature. Like in horizontal geodesics, when $d(u, v)$ is greater than $d_{max}$, we do not generate the geodesic representation for the pair and set their distance to infinity.

### 3.3 Task specific geodesic representation

Task-specific geodesic representation is a collection of pair-wise geodesic representations. Here, we focus on three levels of tasks, node-level, edge-level, and graph-level.

For **node-level** tasks, the goal is to generate a representation $\boldsymbol{z}_v^{(n)}$ for a target node $v$. We first find $v$'s k-hop neighbor nodes $S_v = \mathcal{N}^k(v)$. Then for each node $s_i$ in $S_v$, we find the pair-wise geodesic, $\boldsymbol{g}_{v,s_i}$. Note that in node level tasks, vertical geodesic only contains the geodesic nodes close to $s_i$, $P_{(v,s_i),s_i}^{(V)}$. This enables fast geodesic extraction, and Theorem 1 shows that with only one side of the vertical geodesics, GDGNN can already distinguish almost all pairs of regular graphs. The geodesic representations of the neighborhood are combined by a pooling layer,

$$\boldsymbol{z}_v^{(n)} = R^{(n)}(\boldsymbol{g}_{(v,s_i)}|s_i \in S_v). \tag{7}$$

While the GNN node embeddings of the neighborhood are not conditioned on $v$, by adding the geodesic information to every neighbor, we inject the conditional information of $v$ into its neighbors. Then the collective information of the neighbors helps distinguish $v$.

For **edge-level** tasks, the goal is to generate an edge embedding $\boldsymbol{z}_{(u,v)}^{(e)}$ for nodes $u$ and $v$. Here, we directly use the pair-wise geodesic for the two nodes, $\boldsymbol{z}_{(u,v)}^{(e)} = \boldsymbol{g}_{(u,v)}$. As pointed out by previous works [58, 53], the advantage of this setup is that, when the downstream task is to rank one link against $N$ other links with one same node $v$, we only need to compute the distance vector from $v$ once and find the geodesic nodes efficiently.

For **graph-level** tasks, we need to generate a graph embedding $\boldsymbol{z}_{\mathcal{G}}^{(g)}$ for $\mathcal{G} = (\mathcal{V}, \mathcal{E})$. We use another pooling layer to summarize the geodesic-augmented representation of every node in the graph,

$$\boldsymbol{z}_{\mathcal{G}}^{(g)} = R^{(g)}(\boldsymbol{z}_v|v \in \mathcal{V}). \tag{8}$$

GDGNN can also be easily generalized to other levels of tasks such as subgraph-level.

### 3.4 Expressive Power of GDGNN

Distinguishing power on graph-level tasks is often used to characterize the expressiveness of a GNN model. For GNN models to be more expressive than the 1-WL test, we can test whether they are able to distinguish regular graphs. In the following theorem, we show that given an injective $R^{(g)}$, the graph embedding $\boldsymbol{z}_{\mathcal{G}}$ generated based on vertical geodesics can distinguish most regular graphs.

**Theorem 1.** *Let the graph pooling layer $R^{(g)}$ be injective given input from a countable space. Consider all pairs of n-sized r-regular graphs, where $3 \leq r < \sqrt{2 \log n}$, $n$ is the number of nodes in the graph. For any small constant $\epsilon > 0$, there exists a GDGNN using vertical geodesic representations with $d_{max} = \lceil (\frac{1}{2} + \epsilon) \frac{\log n}{\log(r-1-\epsilon)} \rceil$ which distinguishes almost all $(1 - o(1))$ such pairs of graphs.*

We include the proof of Theorem 1 in Appendix A. This theorem implies that GDGNN with vertical geodesics is more expressive than basic GNNs (and 1-WL) and can discriminate almost all $r$-regular graphs. Moreover, GDGNN only needs to search a small neighborhood for geodesics to achieve higher expressiveness. This guarantees the good efficiency and performance of GDGNN.

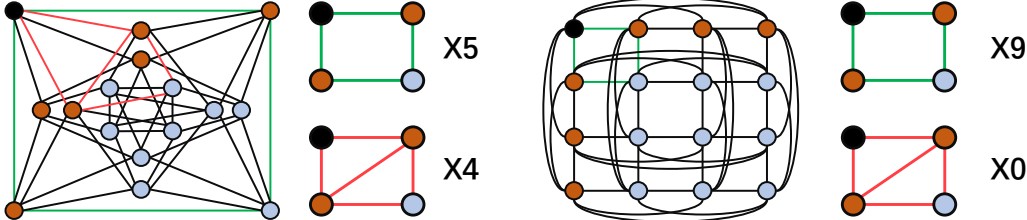

Figure 4: An example of two graphs where GDGNN with vertical geodesics and subgraph degrees can distinguish, while distance encoding cannot.

Moreover, when the vertical geodesic pooling layer encodes the subgraph induced by the geodesic nodes (using Equation 6 for geodesic pooling), GDGNN can distinguish some distance regular graphs. Figure 4 shows two distance regular graphs, the Shrikhande graph, and the 4x4 rook's graph. They have the same intersection array and number of nodes. As proved by Li *et al.* [25], distance encoding is not able to distinguish this pair of graphs, because nodes in the two graphs will be assigned the same embedding. However, a vertical geodesic can discriminate them. Suppose the black nodes are the target nodes, the brown nodes are one distance away from the black nodes and the light blue nodes are two distances away from the black nodes. Consider the light blue nodes in both graphs, they each have a pair of brown common neighbors with a black node. The brown pair is the vertical geodesics of one light blue node. If we count the number of edges in the subgraph formed by each brown pair, we will see that four out of nine pairs form a one-edge graph (in red) in the Shrikhande graph, while zero out of nine pairs form a one-edge graph in the 4x4 rook's graph. Since Equation 6 directly captures this and will give four light blue nodes in the Shrikhande graph different pair-wise geodesics with respect to the black node, an injective graph pooling layer will yield different node representations for the two black nodes, and in turn, discriminates the two graphs.

**Complexity.** The main computation advantage of GDGNN comes from the fact that, for GDGNN, the number of GNN runs stays at a constant of 1, while subgraph-based methods incur GNN runs which increase linearly with respect to the number of queries. Hence, GDGNN maintains the good efficiency of basic GNN methods, while injecting the high-expressiveness of subgraph methods into the model. A detailed complexity analysis on different tasks can be found in Appendix B.

**Limitations.** While a simple geodesic pooling layer satisfies the theoretical expressiveness, if we indeed use a GNN or other more complex layer for pooling as suggested in Section 3.2.2, the pooling will dominate the good amortized complexity of GDGNN. We see this as a possible future direction where we can use a more complex pooling layer to further improve expressiveness while maintaining efficiency. Like subgraph GNNs, the geodesic extraction process also includes a cutoff distance $d_{max}$ which potentially limits the expressive power of GDGNN. Meanwhile, we compare GDGNN with different cutoff distances in Appendix F and show that a relatively small $d_{max}$ suffices to represent the graph structure as indicated in Theorem 1.

## 4 Related Works

**GNNs beyond 1-WL test.** Designing GNNs with stronger representation power is a fundamental task in GNN research. Xu *et al.* [45] first proved that the expressiveness of basic GNNs, such as GCN [20], GraphSAGE [17], and GAT [40], is upper-bounded by the 1-WL test. Following works conducted extensive research to break this limit. They can be sorted into several directions: (1) Earlier works proposed higher-order GNNs that are as expressive as the k-WL algorithm, such as k-GNN [23], RingGNN[6], and PPGN[26]. These methods require message passing on node tuples that grow exponentially with respect to k. The increased computation cost makes them not scalable to large graphs. (2) Another type of methods, like GSN [5] and MotifNet [27], incorporates isomorphism counting on certain graph substructures like cycles, triangles, and cliques, similar to earlier kernel-based methods [46, 12]. They essentially augment the models with features undetectable by GNNs. However, a properly-design substructure usually requires expert knowledge which is not available for all data. (3) Some other works use identity or random node labeling within the plain GNN framework. ID-GNN [49] extracts a subgraph around the target node and performs heterogeneous message passing. DE-GNN [25] also requires subgraph extraction but labels nodes in the subgraph with distance to the targets. They both have stronger expressiveness than basic GNNs. Abboud *et al.*

[1] shows that GNNs possess universality with random node feature initialization (RNI). Nevertheless, RNI also induces a huge sample space, and GNNs adopting RNI is very hard to train. (4) Some recent works propose to use additional subgraph structures to augment the node embeddings. GraphSNN [43] incorporates properties of overlapping subgraphs between adjacent nodes to be more expressive than 1-WL. Nested-GNN [52] applies GNN to ego-subgraphs of every node and proves that it is able to discriminate regular graphs. Sharing the same idea of Nested-GNN, GNN-AK[55] uses a different subgraph pooling strategy and uses subgraph dropping to improve the model scalability. (5) Another approach learns from both positional and structural representation. Dwivedi *et al.* [14] proposes to learn from positional node embeddings unrecognized by the GNN and iteratively update them during message passing. Zhu *et al.* [57] computes node proximity matrix to learn positional of nodes. They can be combined with GDGNN to adapt to position-sensitive tasks.

**Link prediction using GNNs.** Earlier works [21, 20, 7, 39] applied basic GNNs to link prediction using an auto-encoder framework, where the GNN serves as the encoder of the nodes, and edges are decoded by their nodes' encoding vectors. Another line of works, including SEAL [50], IGMC [51], and GraIL [36], use GNNs to encode a labelled subgraph to form the link representation. Later work [34, 54] reveals that a good node-level/graph-level representation does not necessarily lead to a good edge-level representation. Zhang *et al.* [54] systematically show that the labeling trick bridges the gap between node-level and edge-level GNN expressiveness. Sharing the spirits of labeling trick, NBFNet [58] extends the generalized Bellman-Ford algorithm to the neural setting and can encode many traditional path-based methods. NBFNet has better scalability when there are multiple links sharing the same node. Some other works also propose to use path information between links to enhance the prediction like horizontal geodesic. Jagvaral *et al.* [19] uses CNN and BiLSTM to summarize the path information, however, their model requires fixed node embedding, while horizontal geodesic is based on graph structure generated from GNN and hence is inductive. Wang *et al.* [42] learns relational path information by training a set of embeddings for all different types of relational paths, while horizontal GDGNN uses GNN to capture the graph structure and pool node embeddings along the geodesic without training relational-path-specific embeddings.

# 5 Experimental Results

## 5.1 Link Prediction

**Datasets.** We use several types of datasets for link prediction. (1) Knowledge Graph (KG) inductive link prediction datasets. Inductive link prediction means that the model is trained on a training graph while tested on a different graph. This requires the model to generalize to unseen entities, which is a property preserved by a limited number of models. We follow the standard inductive split of WN18RR[10] and FB15K237[38] as in Teru *et al.* [36]. We rank each positive link against 50 randomly sampled negative links with the same head/tail as the positive link and report the Hit@10 ratio. (2) OGB large-scale link prediction dataset [18], including OGBL-COLLAB and OGBL-PPA. We use the official data split. We rank each positive link against a set of provided negative links in the dataset and report the Hit@50 ratio for the OGBL-COLLAB dataset and the Hit@100 ratio for the OGBL-PPA dataset. (4) Transductive KG results are included in Appendix C. (3) Results of citation datasets and comparison to other distance-based methods can be found in Appendix D. These datasets vary by graph size and demonstrate GDGNN's capability across scales.

**Baselines.** We compare GDGNN with Rules Mining methods, DRUM [31] and NeuralLP [47], and GNN-based methods, NBFNet [58] and GraIL [36] for the KG tasks. Some other works like [41, 8] incorporate text data in the KG to achieve better performance. As we believe models that employ textual information is beyond the context of graph structural learning, we exclude them in our comparison. For the OGB dataset, we compare GDGNNs with other methods achieving top places on the OGB link prediction leaderboard, including Deep Walk [29], Resource Allocation (RA) [56], SEAL, Deeper GCN [24].

**Implementation Details.**[1] For link prediction, we use GCN (RGCN for KG) as the basic GNN. For the KG datasets, we search the number of GNN layers in $\{2, 3, 4, 5\}$ and use 64 as hidden dimensions, and the max search distance for geodesic, $d_{max}$, is the same as the number of GNN layers. For the OGB datasets, we search the number of GNN layers in $\{2, 3, 4\}$ and use 100 as the hidden dimension.

---

[1]The code and data of GDGNN can be found at https://github.com/woodcutter1998/gdgnn.

Table 1: Link prediction results (%) on KG.

| Method | FB15K-237 | | | | WN18RR | | | |
|---|---|---|---|---|---|---|---|---|
| | v1 | v2 | v3 | v4 | v1 | v2 | v3 | v4 |
| DRUM | 52.9 | 58.7 | 52.9 | 55.9 | 74.4 | 68.9 | 46.2 | 67.1 |
| NeuralLP | 52.9 | 58.7 | 52.9 | 55.9 | 74.4 | 68.9 | 46.2 | 67.1 |
| GraIL | 64.2 | 81.2 | 82.8 | 89.3 | 82.5 | 78.7 | 58.4 | 73.4 |
| NBFNet | 83.4 | 94.9 | 95.1 | 96.0 | 94.8 | **90.5** | 89.3 | **89.0** |
| GDGNN-Vert | **85.4** | 95.6 | **97.9** | **97.8** | 93.1 | 88.9 | 88.1 | 85.6 |
| GDGNN-Hor | 82.1 | **95.8** | 94.0 | 97.7 | **94.9** | 87.8 | **89.5** | 88.5 |

Table 2: Link prediction results (%) on OGB.

| Method | OGBL-COLLAB | OGBL-PPA |
|---|---|---|
| GCN | $44.75 \pm 1.45$ | $18.67 \pm 1.32$ |
| DEEP WALK | $50.37 \pm 0.34$ | $28.88 \pm 1.63$ |
| RA | - | $\mathbf{49.33} \pm 0.00$ |
| DeeperGCN | $52.73 \pm 0.47$ | - |
| SEAL | $54.37 \pm 0.49$ | $48.80 \pm 3.16$ |
| GDGNN-Vert | $\mathbf{54.74} \pm 0.48$ | $45.92 \pm 2.14$ |
| GDGNN-Hor | $54.52 \pm 0.72$ | $28.79 \pm 3.80$ |

Table 3: Link prediction running time results.

| Method | OGBL-COLLAB | OGBL-PPA |
|---|---|---|
| GCN | 27 sec | 17 min |
| SEAL | 90 sec | 2 hr 20 min |
| NBFNet | OOM | OOM |
| GDGNN-Vert | 36 sec | 33 min |

Table 4: Graph classification running time results.

| Method | OGBG-MOLHIV | OGBG-MOLPCBA |
|---|---|---|
| GIN | 20 sec | 2 min |
| GIN-AK | 70 sec | 12 min |
| Nested-GIN | 149 sec | 16 min |
| GDGNN-Vert | 25 sec | 2.5 min |

We train 50 epochs with a batch size of 64 for the KG datasets, and we train 25 epochs with a batch size of 2048 for the OGB datasets. For OGBL-COLLAB dataset, we use homogeneous labels (**1** labels) for all nodes. We record the test score using the model with the best set of hyperparameters on validation set. The experiment is repeated 5 times and we take the average.

**Results and discussion.** We achieved SOTA performance on the inductive datasets. GDGNN is able to significantly improve over rule-mining methods and GraIL. We also outperform NBFNet on most of the datasets. On the OGBL-COLLAB dataset (Table 2), GDGNN achieved SOTA performance. On the OGBL-PPA dataset, vertical GDGNN can obtain a very competitive result. Horizontal GDGNN does not perform as well possibly because a single path has limited distinguishing power in a graph with high density. On both OGB datasets, GDGNN improves greatly over its basic GNN, GCN. Moreover, as shown in Table 3, GDGNN has running-time superiority over methods achieving comparable performance and has a memory advantage to handle large graphs where NBFNet cannot when links do not share common nodes. To make a fair running time comparison, we run all models on 32 CPUs and 1 Nvidia GeForce 1080Ti GPU. We also include ablation studies on all components of GDGNN in Appendix E.

## 5.2 Graph Classification

**Datasets.** We use two types of datasets for graph classification. (1) TU datasets contain D&D[11], MUTAG[9], PROTEINS[11], PTC_MR[37]. The evaluation metric is Accuracy (%). We follow the evaluation protocol in [52], where we use 10-fold cross-validation to compute the score. (2) OGB datasets[18], including OGBG-MOLHIV and OGBG-MOLPCBA. The task of the OGBG-MOLHIV dataset is to predict whether a molecule inhibits the HIV virus. We use Area Under ROC Curve (AUC) to evaluate the OGBG-MOLHIV dataset. OGBG-MOLPCBA dataset has 128 classification tasks. We follow the evaluation protocol provided by the OGB team and use Average Precision (AP) averaged across all classes to evaluate the dataset. (3) Synthetic datasets, including EXP[1] and CSL[28]. These datasets contain graphs that can be only distinguished by models with $> 1$-WL expressiveness. We report accuracy for these datasets.

**Baselines.** For the TU dataset, we compare GDGNN with another plug-and-play GNN framework, Nested-GNN[52]. Both GDGNN and Nested-GNN use GIN and GCN as their basic GNNs. For the OGB dataset, we use GIN as the basic GNN, we compare GDGNN with methods achieving top places on the OGB leaderboard, including GIN[45], GIN-AK [55], LSPE [14], Nested GNN [52], Directional GSN (DGSN) [3], PF-GNN [13].

**Implementation details.** We use vertical geodesic in the graph-level tasks because of its theoretical advantage over horizontal GDGNN. For TU datasets, we search the GNN layers and $d_{max}$ in {2,3,4}. We train on the TU dataset for 100 epochs with a batch size of 64. For OGB datasets, we search the number of GNN layers in {2,3,4,5}, and train for 100 epochs with a batch size of 256. We adopt the virtual node technique [16] to increase the connectivity of graphs for the OGBG-MOLPCBA dataset. For synthetic datasets, we search the number of GNN layers in {2,3,4}, and train for 50 epochs with a batch size of 16. For the synthetic datasets, we report GDGNN's performance with different numbers of layers {2,3,4}. We train the models for 100 epochs with a batch size of 8.

Table 5: Graph classification results (%) on TU.

| | D&D | MUTAG | PROTEINS | PTC_MR |
|---|---|---|---|---|
| Avg. #nodes | 284.32 | 17.93 | 39.06 | 14.29 |
| GCN | $71.6 \pm 2.8$ | $73.4 \pm 10.8$ | $71.7 \pm 4.7$ | $56.4 \pm 7.1$ |
| GIN | $71.6 \pm 3.0$ | $74.0 \pm 8.8$ | $71.2 \pm 5.2$ | $57.0 \pm 5.5$ |
| Nested-GCN | $76.3 \pm 3.8$ | $82.9 \pm 11.1$ | $73.3 \pm 4.0$ | $57.3 \pm 7.7$ |
| Nested-GIN | $\mathbf{77.8} \pm 3.9$ | $87.9 \pm 8.2$ | $\mathbf{73.9} \pm 5.1$ | $54.1 \pm 7.7$ |
| GDGCN | $77.6 \pm 4.0$ | $88.4 \pm 6.6$ | $73.7 \pm 3.4$ | $\mathbf{60.3} \pm 4.5$ |
| GDGIN | $\mathbf{77.8} \pm 3.6$ | $\mathbf{89.4} \pm 7.1$ | $73.6 \pm 2.5$ | $57.9 \pm 3.4$ |

Table 6: Graph classification results (%) on OGB.

| Method | OGBG-MOLHIV | OGBG-MOLPCBA |
|---|---|---|
| GIN | $75.58 \pm 1.40$ | $27.03 \pm 0.23$ |
| LSPE | - | $28.40 \pm 0.20$ |
| Nested-GIN | $78.34 \pm 1.86$ | $28.32 \pm 0.41$ |
| DGSN | $\mathbf{80.39} \pm 0.90$ | - |
| GIN-AK | $78.22 \pm 0.75$ | $\mathbf{29.30} \pm 0.44$ |
| PF-GNN | $80.15 \pm 0.68$ | - |
| GDGIN | $79.07 \pm 1.20$ | $28.59 \pm 0.63$ |

**Results and discussions.** On the TU datasets (Table 5), GDGNNs consistently outperform the basic GNNs. Compared to Nested-GNN, GDGNN is able to bring a larger performance increase. This shows GDGNN's enhancing power. On the OGB datasets (Table 6), we can see that GDGNN's performance is lower than PF-GNN and Directional GSN (so are GIN-AK and Nested-GNN). This might indicate that pairwise conditional information is less effective for graph tasks than for link tasks.

Nevertheless, GDGNN still outperforms the basic GIN a lot, and as GDGNN is a general framework that can be applied to any basic GNN, the performance can be further improved. We notice that GDGNN outperforms LSPE on the OGB-MOLPCBA dataset, which might indicate that the pair-wise relation encoded by GDGNN is sufficient in representing the positional information of a node. On the synthetic data, we achieve comparable results to the universal PF-GNN, which verifies our theory. We include ablation studies of graph-level geodesics in Appendix E.

Table 7: Synthetic Results.

| | EXP | CSL |
|---|---|---|
| GIN | 50.0 | 10.0 |
| RNI | 99.7 | 16.0 |
| PF-GNN | 100.0 | 100.0 |
| 2-GDGNN | 50.0 | 30.0 |
| 3-GDGNN | 100.0 | 60.0 |
| 4-GDGNN | 100.0 | 100.0 |

### 5.3 Node Classification

**Datasets.** We use airport datasets, Brazil-Airport, Europe-Airport, and USA-Airport for our node classification experiments. The task is to predict passenger flow level from the flight traffic network, where graph structure matters and expressive GNNs are useful. Following the setting in Li *et al.* [25], we split the dataset with a train/test/valid ratio of 8:1:1 and run experiments with independent random initialization 20 times and report the average accuracy and 95% confidence range.

**Baselines.** For GNN baselines, we compare to 1-WL GNNs, including GCN and GIN, and DE-GNN [25] that also relies on distance information. We further compare to Struc2vec[30], PhUSION[57] that encodes graph structures without GNN. DE-GNN and PhUSION have several variants and we take their best results here. Like in graph classification, we use vertical geodesics because of its good theoretical properties. We search the GNN layers and $d_{max}$ in {2,3,4}. We train on all datasets for 200 epochs with a batch size of 32.

Table 8: Node classification results (%).

| Method | Bra.-Airports | Eur.-Airports | USA-Airports |
|---|---|---|---|
| GCN | $64.55 \pm 4.18$ | $52.07 \pm 2.79$ | $56.58 \pm 1.11$ |
| GIN | $72.83 \pm 3.57$ | $53.84 \pm 3.94$ | $57.43 \pm 2.13$ |
| PhUSION | $72.37 \pm 0.00$ | $56.02 \pm 0.00$ | $63.49 \pm 0.00$ |
| Struc2vec | $70.88 \pm 4.26$ | $57.94 \pm 4.01$ | $61.92 \pm 2.61$ |
| DE-GNN | $75.37 \pm 3.25$ | $\mathbf{58.41} \pm 3.20$ | $64.16 \pm 1.70$ |
| GDGIN | $\mathbf{78.61} \pm 2.20$ | $53.97 \pm 3.50$ | $\mathbf{64.36} \pm 1.62$ |

**Results and discussions.** From Table 8, we can see that GDGNN achieves very competitive results compared to DE-GNN, and significantly improves the performance of basic GIN on two datasets, which experimentally verifies our claim that node-level geodesic is more expressive than plain GNNs. For GDGNN, because the GNN is applied to the graph only once in the entire node classification process, it can maintain a good efficiency comparable to that of the basic GNNs.

## 6 Conclusions

In this paper, we propose a general GNN framework that incorporates geodesic information to generate node, link, and graph representations. Our framework is more powerful than basic GNNs both practically and theoretically and is more efficient than current more-expressive GNNs as demonstrated by the running-time experiments. Our framework has the potential to make GNNs encoding conditional information really applicable to large-scale real-world graphs.

**Acknowledgement.** LK and YC are supported by NSF grant CBE-2225809. MZ is supported by NSF China (No.62276003) and CCF-Baidu Open Fund (NO.2021PP15002000).

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
