# Appendix

## A    Proof of Theorem 1

The proof of Theorem 1 generally follows the proof of Nested-GNN's expressiveness[52]. Let $Q_{v,\mathcal{G}}^k$ be the set of nodes in $\mathcal{G}$ that are exactly $k$ distance away from node $v$. We can have the following definition,

**Definition A.1.** *The edge configuration bewteen $Q_{v,\mathcal{G}}^k$ and $Q_{v,\mathcal{G}}^{k+1}$ is a list $C_{v,\mathcal{G}}^k = (a_{v,\mathcal{G}}^{1,k}, a_{v,\mathcal{G}}^{2,k}, ...)$ where $a_{v,\mathcal{G}}^{i,k}$ denotes the number of nodes in $Q_{v,\mathcal{G}}^{k+1}$ of which each has exactly $i$ edges from $Q_{v,\mathcal{G}}^k$.*

The proof leverages the fact that nodes in randomly sampled regular graphs are very likely to have different edge configurations within their small neighborhood, and GDGNN directly captures this distinction to discriminate the graphs. Formally, we have the following excerpted Lemma from [52],

**Lemma 2.** *For two graphs $\mathcal{G}^{(1)} = (\mathcal{V}^{(1)}, \mathcal{E}^{(1)})$ and $\mathcal{G}^{(2)} = (\mathcal{V}^{(2)}, \mathcal{E}^{(2)})$ that are uniformly independently sampled from all n-node r-regular graphs, where $3 \leq r < \sqrt{2 \log n}$, we pick any two nodes, each from one graph, denoted by $v_1$ and $v_2$ respectively. Then there is at least one $i \in (\frac{1}{2} \frac{\log n}{\log(r-1-\epsilon)}, (\frac{1}{2} + \epsilon)\frac{\log n}{\log(r-1-\epsilon)})$ with probability $1 - o(n^{-1})$ such that $C_{v_1, \mathcal{G}^{(1)}}^i \neq C_{v_2, \mathcal{G}^{(2)}}^i$.*

*Proof.* To prove Theorem 1, we first assume, without loss of generality, that the hidden dimension of GDGNN is one. We know plain message passing GNN will generate the same embeddings for every node in the regular graph because all nodes have the same degree/subtree. Hence, we can normalize all the node representations to a constant of one. For simplicity, we also reduce vertical geodesic pooling function $R_{gd}^{(V)}$ in Equation 4 to a simple sum of the representations of the geodesic nodes.

Let $d_{max} = \lceil (\frac{1}{2} + \epsilon)\frac{\log n}{\log(r-1-\epsilon)} \rceil$, the maximum distance in which GDGNN searches for a target node's neighbors. We then consider node $v_i$ and it's $d_{max}$-hop rooted subgraph $\mathcal{G}_{v_i}^{d_{max}}$, the vertical geodesic pooling process is essentially counting the number of geodesic nodes of each node $v_j$ in $\mathcal{G}_{v_i}$. And the pair-wise vertical geodesic between the $v_i$ and $v_j$ becomes,

$$\boldsymbol{g}_{v_i, v_j}^{(V)} = |\{P_{v_j}^{(V)}\}| \oplus d(v_i, v_j) \tag{9}$$

where $d(v_i, v_j)$ is the distance between $v_i$ and $v_j$, and hence $v_j \in Q_{v_i, \mathcal{G}_{v_i}}^{d(v_i, v_j)}$ and $\boldsymbol{g}_{v_i, v_j}^{(V)}$ records the number of edges of $v_j$ that connects to nodes in $Q_{v_i, \mathcal{G}_{v_i}}^{d(v_i, v_j)-1}$, and its distance to $v_i$. We also have the node-level pooling function to be a distance sorted bin count function on the set of pair-wise vertical geodesic representations. It first sorts the geodesic representations by their second dimension (distance) and performs bin count on their first dimension (number of geodesic nodes) separately for each distance. Then GDGNN produces an mapping of $\mathcal{G}_{v_i}$ to its exact set of edge configurations, $\mathcal{C}_{v_i} = (C_{v_i, \mathcal{G}_{v_i}}^1, C_{v_i, \mathcal{G}_{v_i}}^2, ...)$.

Let $\mathcal{G}^{(1)} = (\mathcal{V}^{(1)}, \mathcal{E}^{(1)})$ and $\mathcal{G}^{(2)} = (\mathcal{V}^{(2)}, \mathcal{E}^{(2)})$ be two graphs uniformly independently sampled from all n-node r-regular graphs. We consider two nodes $v_i \in \mathcal{V}^{(1)}$ and $v_j \in \mathcal{V}^{(2)}$. Because $d_{max} = \lceil (\frac{1}{2} + \epsilon)\frac{\log n}{\log(r-1-\epsilon)} \rceil$, the sets of edge configurations of the rooted subgraphs, $\mathcal{C}_{v_i}$ and $\mathcal{C}_{v_j}$ are different with probability $1 - o(n^{-1})$, according to Lemma 2. In such case, GDGNN also maps their rooted subgraphs to different edge configurations, and hence the node-level geodesic representation $\boldsymbol{z}_{v_i}^{(n)}$ and $\boldsymbol{z}_{v_j}^{(n)}$ generated by GN-GNN will be different with the same probability.

Consider $v_i$ and all node $v_j \in \mathcal{V}^{(2)}$, by union bound, $\boldsymbol{z}_{v_i} \notin \{\boldsymbol{z}_{v_j} | v_j \in \mathcal{V}^{(2)}\}$ with a probability of $1 - o(1)$. □

## B    Complexity Analysis

The complexity of GDGNN differs by task. On almost all tasks, GDGNN demonstrates superiority in efficiency. The complexity of GDGNN can be divided into two parts, the geodesic extraction part, and the GNN part. We first note that the complexity of geodesic extraction and GNN cannot be directly compared as their computation units are different, geodesic extraction operates on unit

integers while GNN operates on $d$-dimensional vectors that are potentially very large, and running a geodesic extraction on a graph takes significantly less amount of time compared to running a GNN on the same graph. Meanwhile, we can always pre-compute distance for fast geodesics extraction during inference. Hence, we present a complexity analysis of GDGNN on the GNN part.

## B.1 Graph and node level tasks

The analysis of graph and node level tasks is similar as graph representation is generated by applying mean/max pooling on all node representations. Specifically, consider the problem where we need to infer $k$ nodes in a graph with $|V|$ nodes and $|E|$ edges. In worst-case, the complexity of a $T$-layer subgraph-based GNN is $O(kT|E|)$, because it applies a $T$-layer GNN on $k$ nodes' subgraphs, where each subgraph contains $|E|$ edges (the full graph). GDGNN's worst-case complexity is $O(T|E| + k|E|)$, $O(T|E|)$ is for applying the GNN once to the full graph, and the number of nodes in a vertical geodesic is bounded by the number of edges and the pooling takes $O(k|E|)$. Hence, GDGNN is more efficient than subgraph GNNs. For graphs, the comparison becomes $O(|V|T|E|)$ versus $O(T|E| + |V||E|) = O(|V||E|)$, this shows that GDGNN helps amortize the expensive computation cost, $T$ versus 1, of applying a GNN.

## B.2 Edge level tasks

For edge-level tasks, we consider the worst-case scenario where we try to predict $k$ links on a graph with $|V|$ nodes and $|E|$ edges. The $k$ links do not share any common nodes. GDGNN takes $O(T|E| + k|V|)$ for $O(T|E|)$ GNN on the full graph and $k$ geodesic pooling, where the number of nodes in a link-level geodesic is bounded by the number of nodes $O(|V|)$ in the graph (for both vertical and horizontal geodesics). Subgraph GNN methods take $O(kT|E|)$ for applying $T$-layer GNN onto $k$ links' subgraphs (worst case $O(|E|)$ edges). NBFNet [58] shares the same complexity because the links do not have common nodes, and the results of one GNN run can not be shared across links as suggested by the authors of NBFNet. Thus, $k$ runs of GNN are necessary for the NBFNet. Then, comparing the complexities of GDGNN and subgraph GNNs, when we have fewer query links and hence $O(T|E|) > O(k|V|)$, the subgraph method's complexity $O(kT|E|)$ grows linearly with respect to the number of queries, while GDGNN's complexity $O(T|E|)$ does not. When we have more query links and $O(T|E|) < O(k|V|)$, GDGNN's complexity $O(k|V|)$ is more optimal than subgraph GNNs' $O(kT|E|)$ complexity.

## C Transductive Knowledge Graph link prediction

For transductive KG link prediction, we follow the inductive setting as in Section 5.1. Following [4], we rank a positive link against all negative links that have the same head (or tail) as the positive links. We additionally compared to general knowledge graph link prediction methods that are not inductive, including TransE [4], RotatE [35], and RGCN [32]. We report H@N and Mean Reciprocal Rank (MRR) for the transductive setting.

Table 9: Knowledge graph transductive link prediction results.

| Method | FB15K-237 | | | | WN18RR | | | |
|---|---|---|---|---|---|---|---|---|
| | MRR | H@1 | H@3 | H@10 | MRR | H@1 | H@3 | H@10 |
| DRUM | 34.3 | 25.5 | 37.8 | 51.6 | 48.6 | 42.5 | 51.3 | 58.6 |
| NeuralLP | 24.0 | - | - | 36.2 | 43.5 | 37.1 | 43.4 | 56.6 |
| TransE | 29.4 | - | - | 46.5 | 22.6 | - | - | 50.1 |
| RotatE | 33.8 | 24.1 | 37.5 | 55.3 | 47.6 | 42.8 | 49.2 | 57.1 |
| RGCN | 27.3 | 18.2 | 30.3 | 45.6 | 40.2 | 34.5 | 43.7 | 49.4 |
| NBFNet | **41.5** | **32.1** | **45.4** | **59.9** | **55.1** | **49.7** | **57.3** | **66.6** |
| GDGNN-Vert | 23.2 | 15.8 | 26.4 | 45.1 | 46.2 | 39.3 | 48.6 | 59.1 |
| GDGNN-Hor | 25.1 | 16.2 | 28.7 | 44.9 | 43.2 | 35.2 | 47.2 | 58.0 |

We acknowledge that GDGNN does not perform as well in the transductive setting as in the inductive setting. However, GDGNN is still able to greatly improve the performance of RGCN on the WN18RR dataset, and RGCN requires trainable node embeddings while GDGNN does not. We suspect that the performance discrepancy between the homogeneous setting and the knowledge graph setting is due to the increased number of edge types. WN18RR dataset has 11 types of relations, and FB15K237 has 237. Note that GNN methods like NBFNet[58] train a set of weights for every target relation type, meaning that their message passing is conditioned on the relation, while GDGNN only uses one set of weights. We notice that GDGNN is able to outperform some embedding methods on the WN18RR dataset, while not performing as good on FB15K237, possibly due to the fact that the FB15K237 dataset contains much more relations than the WN18RR dataset. This aligns with our hypothesis. A potential solution is to also train multiple GDGNNs that each handle one target relation type, note that we still only need to run each relation-specific GDGNN once and keep the good amortized property of GDGNN. We leave this to future work.

## D   More link prediction results

Citation datasets[33] include Cora, Citeseer, and Pubmed datasets. We follow the experimental setting in [58] and use 85:5:10 for the train/valid/test links split. The evaluation metric is ROC-AUC (AUC) and Average Precision (AP) scores. We compare GDGNN with popular link prediction methods VGAE [21], SEAL[50], NBFNet[58] for the citation datasets. Note that for citation data, we do not use the pre-trained node embeddings from the publications' content, incorporating such information only increases the power of GDGNN.

We use GCN as the basic GNN. We search the number of GNN layers in $\{2, 3, 4, 5\}$, and the max search distance for geodesic, $d_{max}$, is the same as the GNN layers. We use 32 hidden dimensions for all fully-connected layers in the model. We train 100 epochs with a batch size of 64 for the citation datasets.

Table 10: Link prediction results (%) on Citation dataset.

| Method | Cora | | Citeseer | | PubMed | |
|---|---|---|---|---|---|---|
| | AUC | AP | AUC | AP | AUC | AP |
| VGAE | 91.40 | 92.60 | 90.80 | 92.00 | 94.40 | 94.70 |
| SEAL | 93.32 | 94.21 | 90.52 | 92.43 | 97.78 | 97.90 |
| NBFNet | **95.61** | **96.17** | **92.28** | 93.74 | **98.30** | 98.15 |
| GDGNN-Vert | 94.47 | 95.71 | 91.98 | **94.01** | 98.16 | **98.17** |
| GDGNN-Hor | 94.56 | 95.48 | 92.06 | 93.59 | 97.83 | 98.10 |

On the citation dataset (Table 10), GDGNN achieved very competitive results compared to the SOTA method NBFNet, and GDGNN is more efficient than NBFNet as demonstrated in Table 3.

We also include C.elegans, NS, and PB datasets adopted by DE-GNN [25]. We follow its experiment setup to split the edges into 8:1:1 train/test/valid splits and repeat the experiment 20 times to report the average AUC and 95% confidence range. For these datasets, we compare with other methods that also rely on distance encoding, including DE-GNN, PGNN[48], SEAL, and the basis GNN, GIN.

Table 11: Link prediction results compared to other distance-related methods.

| Method | C.elegans | NS | PB |
|---|---|---|---|
| GIN | $75.58 \pm 0.59$ | $87.75 \pm 0.56$ | $91.11 \pm 0.52$ |
| PGNN | $78.20 \pm 0.33$ | $94.88 \pm 0.77$ | $89.72 \pm 0.32$ |
| SEAL | $88.26 \pm 0.56$ | $98.55 \pm 0.32$ | $94.18 \pm 0.57$ |
| DE-GNN | $\mathbf{90.05} \pm 0.26$ | $\mathbf{99.43} \pm 0.63$ | $94.95 \pm 0.37$ |
| GDGNN-Vert | $87.88 \pm 0.42$ | $98.10 \pm 0.26$ | $94.43 \pm 0.39$ |
| GDGNN-Hor | $89.83 \pm 0.70$ | $98.65 \pm 0.48$ | $\mathbf{96.14} \pm 0.73$ |

From the experiment, we can see that GDGNN achieves very competitive results compared to distance-encoding GNN and SEAL, and outperforms PGNN[48] significantly on all datasets. We

suspect the key reason is that PGNN relies on relative positions to the anchors, but the anchors that PGNN randomly chooses are not necessarily representative to all links. In contrast, the geodesic information is directly associated with each link.

# E    Ablation study

GDGNN has many components, and here we present an ablation study of GDGNN to demonstrate the effect of each component. For link prediction datasets, we consider 4 settings, plain GCN (GCN), GCN and distance between the two target nodes of the link (GCN+Dist), GDGNN-Vertical without the geodesic degree (GDGNN-Vert), and GDGNN-Vertical with geodesics node degree (GDGNN-Vert-Deg). Note that the results we report in Section 5 include geodesic node degree.

Table 12: Ablation study on link prediction datasets.

| Method | Cora (AUC) | OGBL-COLLAB (H@50) | OGBL-PPA (H@100) |
|---|---|---|---|
| GCN | 81.79 | 44.75 | 18.67 |
| GCN+DIST | 92.93 | 53.82 | 20.39 |
| GDGNN-Vert | 94.14 | 54.38 | 43.86 |
| GDGNN-Vert-Deg | 94.56 | 54.74 | 45.92 |

From the experiment results (Table 12), we can see that distance is already able to assist link prediction, especially on sparser datasets like Cora and OGBL-COLLAB. However, on OGBL-PPA, where the connectivity of the graph is much larger, distance does not help, whereas geodesic representation is able to significantly improve the expressive power of basic GNNs. Also, when the graph is sparse, it is rather unlikely for the geodesic nodes to form subgraphs, meaning that most of the nodes on the vertical geodesic have a degree of zero, and hence geodesic degree does not improve much. Meanwhile, in OBGL-PPA, geodesic degrees are able to increase the performance of GDGNN, which aligns with the intuition and example we present.

For horizontal geodesics, we present results where only part of the horizontal geodesic is used.

Table 13: Partial horizontal geodesic results.

| Method | Cora | OGBL-COLLAB |
|---|---|---|
| 4-GDGNN-Hor | 94.40 | 54.21 |
| 4-GDGNN-Partial | 92.73 | 53.17 |
| 5-GDGNN-Hor | 94.37 | 53.84 |
| 5-GDGNN-Partial | 92.68 | 53.29 |

Partial represents geodesics where only the head/tail, the head/tail's direct neighbors on the shortest path are used to form the geodesic. N-GDGNN means the cutoff distance is $N$. We do not compare with 2-3 distances because Partial and horizontal geodesic will be exactly the same in that case. We add distance to both Partial and horizontal geodesics to make a fair comparison. We can see that without the full horizontal geodesic, the performance on the Cora dataset indeed dropped by 1.5%. While the difference is not as significant, we still see that horizontal geodesics outperforms Partial on the OGBL-COLLAB dataset. This shows that while being connected within some distance is already a good indicator of the likelihood of the link, incorporating the node structure information can still improve the representation of the link.

For graph classification tasks, we study two simpler versions of GDGNN. We reduce the vertical pair-wise geodesic function (Equation 4) to be a pooling function only on the embeddings of the pair of nodes. Then, the node-level geodesic is essentially combining the node embedding and its k-hop neighbor embedding, we refer to this as Nei. Another version is we append the pair-wise distance to the neighbor embeddings, we refer to this as Dist. Note that vertical geodesic takes part of the direct neighbors of a neighbor node, we also study the variant where all neighbors are used as the geodesic, we call this FullGDNei.

Table 14: Ablation study on graph classification datasets.

| Method | MUTAG (Accuracy) | PROTEINS (Accuracy) | OGBG-MOLHIV (AUC) |
|---|---|---|---|
| GIN | 74.0 | 71.2 | 75.5 |
| GIN+Nei | 81.2 | 71.9 | 76.2 |
| GIN+Nei+Dist | 88.1 | 71.8 | 76.0 |
| GIN+FullGDNei+Dist | 88.1 | 71.9 | 76.4 |
| GDGNN-Vert | 89.0 | 73.3 | 77.9 |
| GDGNN-Vert-Deg | 89.4 | 73.6 | 79.1 |

From Table 14, we can see that the neighbor embedding alone is not very effective, and we need to assist it with pair-wise distance, an explicit form of geodesic to improve the performance. By incorporating the geodesic representation, we are able to consistently improve over the 'Nei+Dist' version, which aligns with Theorem 1. Geodesic degrees do not give significant improvement on the TU dataset possibly due to its high testing variance, but we can still see a notable improvement in the OGBG-MOLHIV dataset. For FullGDNei, we see that it does not bring much performance increase onto GIN+Nei+Dist compared to GDGNN-Vert. This shows that FullGDNei essentially weighs in as an extra layer of GNN which can be covered by hyperparameter search.

## F   Impact of different cutoff distances

A non-negligible hyperparameter in our model is the cutoff distance. In this section, we present results on the impact of different cutoff distances.

Table 15: Performance on link and graph datasets with different cutoff distances (%).

| Method | Cora(L) | OGBL-PPA(L) | OGBG-MOLHIV(G) |
|---|---|---|---|
| 1-GDGNN | 82.46 | 21.15 | 75.68 |
| 2-GDGNN | 91.58 | 43.76 | 78.13 |
| 3-GDGNN | 93.62 | 45.92 | 79.07 |
| 4-GDGNN | 94.47 | 44.82 | 78.84 |

(L) represents the link prediction task, and (G) represents the graph classification task. N-GDGNN means GNN with different max-cutoff distances, we use vertical geodesics with 3-layers of base GNN. From Table 15, we see that for link prediction tasks (Cora in particular), as the cutoff distance increases, the performance increases, this is because more links can be connected by geodesic extraction, and distance itself is already a good indicator of the likelihood of the link. However, disconnected nodes are not always predicted as negative. When disconnected, the link will have zero geodesic representation, but **still have meaningful node representations from the base GNN**. In such a case, GDGNN degenerates embeddings similar to models like VGAE, which is still able to statistically learn the probability of a link based on the node structure around the two nodes of the link. The choice of cutoff distance is data-dependent, as we can see in the results of OGBL-PPA and OGBG-MOLHIV, 4-GDGNN is worse than 3-GDGNN, and the actual number can be determined by hyperparameter tuning. In general, cutoff distance resembles the max number of hops in the subgraph extraction process.

## G   Limitation of Horizontal Geodesics

Figure 5 shows an example where vertical geodesic is more expressive than horizontal geodesic. The gray nodes are symmetric and hence will be assigned the same embedding by a 1-WL GNN, the yellow nodes, because they all connect to two gray nodes, will also be assigned the same embedding by a 1-WL GNN. In this case, both horizontal geodesics between AB and BC are gray-yellow-gray nodes, hence we can not differentiate the two links. Nevertheless, the vertical geodesic of AB is one yellow node, while the vertical geodesic of BC is three

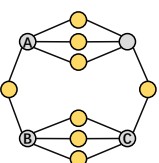

Figure 5

yellow nodes, and the two links will have different representations after summarizing the vertical geodesic information.