# OpenReview forum: "Geodesic Graph Neural Network for Efficient Graph Representation Learning"
_NeurIPS.cc/2022/Conference — NeurIPS 2022 Accept_

### Official Review · Reviewer_wYEs · 2022-07-08

**Rating:** 7
**Confidence:** 5
**Soundness:** 4 excellent
**Presentation:** 3 good
**Contribution:** 3 good

**Summary:**

Among existing graph neural networks (GNNs), standard GNNs are efficient but cannot distinguish structures beyond 1-WL test, while subgraph extraction or customized labeling methods are powerful but require much more computation. This paper proposes a new graph neural network (GNN) architecture, which seeks for a new sweet point between efficiency and effectiveness. The idea is to first compute node representations with a standard GNN, and then readout augmented node/edge/graph-level representations via geodesics. The authors propose two variants for Geodesic GNN, namely horizontal and vertial geodesics. Experiments on both link prediction and graph classification tasks show that geodesic GNNs achieve results on par with SotA models, while only take time similar to standard GNNs.

The contributions of this paper include:

1. Propose geodesic GNNs that are both effcient and effective.
2. Theoretically prove geodesic GNNs are more powerful than 1-WL test.
3. Empirically show that geodesic GNNs achieve good results in both performance and time.

**Questions:**

- Line 55: Why are higher-order GNNs at least cubic complexity? Could you provide a reference to that?
- Line 125: Taking link prediction as an example
- Line 151: Why is the shortest path viewed as a chain of nodes with embeddings one? What are the embeddings discussed here?
- Line 179: You may provide a motivating example here.
- Line 198: If the pooling layer is a small plain GNN, is geodesic GNNs as slow as SEAL?
- Line 214: Why are geodesic nodes close to $v$ ignored here?
- Line 220-224: To my understanding, this property has been discussed in NBFNet[1] and RED-GNN[2] before. It’s better to cite them to provide a fair context.
- Line 297-299: If I understand correctly, the main advantage of GD-GNN over NBFNet is that it avoids the cost of GNNs in subgraph extraction, but GD-GNN still needs to perform subgraph extraction for each query. Why is GD-GNN efficient in any scenario? Is it over-claiming?
- Appendix B: The time complexity analysis is too optimistic. The authors seem to use the average degree of a node in the full graph / in a local subgraph interchangeably, and assume they are small constants. This may not be true. For example, for a scale-free graph, the average degree may be a constant, but the average degree in a local subgraph may not! This is because every node is likely to be connected to nodes with large degrees, and therefore a lot of subgraphs contain nodes with large degrees. I suggest including the complexity of standard GNNs, SEAL, NBFNet for a fair context.
- Table 3: Why does NBFNet cause OOM? Could you give an explanation?

[1] Zhaocheng Zhu et al. Neural bellman-ford networks: A general graph neural network framework for link prediction. NeurIPS 2021.

[2] Yongqi Zhang, and Quanming Yao. Knowledge graph reasoning with relational digraph. WWW 2022.

**Limitations:**

The authors don’t discuss the limitations and societal impacts in the paper. I would suggest the authors adding one paragraph to discuss that. For example, some limitations could be that GD-GNN can only be applied with simple geodesic readout, otherwise it may still have time complexity issues.

**Strengths And Weaknesses:**

Strength

- The idea of using geodesics for readout seems novel to the community. It fills an important gap in GNNs beyond 1-WL test.
- The proposed geodesic GNNs are more powerful in theory, and trades off efficiency and effectiveness better in experiments.
- Geodesic GNNs are general and have the potential to be applied to most standard GNNs.
- The paper is well written and easy to follow.

Weakness

- Although the authors propose node-level representations for geodesic GNNs, there is no experiments on node classification tasks to verify its efficiency and effectiveness.
- The authors are ambiguous about the time complexity of geodesic GNNs. In general, the complexity of geodesic GNNs should between standard GNNs and subgraph extraction. The condition for “GD-GNN maintains the good efficiency of basic GNN models” is that the geodesic readout can be amortized by the cost of the GNN. This is only true for some simple geodesic readout functions, like the two proposed in the paper. The authors should make this more explicit to the readers. Also the time complexity in Appendix B is too optimistic.

---

> ### Author Response · Authors · 2022-08-02
> **Author Response 1/3**
>
> We thank reviewer wYEs for their insightful comments. We address the concerns and questions in the review as follows.
>
> - Re 1 : "Although the authors propose node-level representations for geodesic GNNs, there is no experiments on node classification tasks to verify its efficiency and effectiveness."
>
> We didn't include the node-level evaluation initially, as graph-level evaluation is representative of node-level tasks. But we agree with reviewer wYEs node-level experiments will give a more comprehensive evaluation of GDGNN, and we present the results as follows.
>
> |            |   Bra.-Airports   |   Eur.-Airports   |    USA-Airports   |
> |------------|:-----------------:|:-----------------:|:-----------------:|
> | DE-GNN     |   75.37±3.25   | **58.41±3.20** |   64.16±1.70   |
> | GIN       |   72.83±3.57   |   53.84±3.94   |   57.43±2.13   |
> | GDGNN-ver | **78.61±2.20** |   53.97±3.50   | **64.36±1.62** |
>
> We compare to DE-GNN [5] on the airport datasets, as DE-GNN also uses distance information. The task is to predict passenger flow levels solely from the flight traffic network. We choose these datasets because the node classes reflect their structural roles in the network instead of community identifiers as in commonly used citation networks (such as Cora and Citeseer). Only in these datasets, graph structure matters, and more expressive GNNs are useful [5]. DE-GNN has several variants and we take their best result here. From the experiment, we can see that GD-GNN achieved very competitive results compared to distance-encoded GNN, and significantly improved the performance of base GIN on two datasets, which experimentally verify our claim that node-level geodesic is more expressive than plain GNNs. The results are included in the revision (Appendix E).
>
> - Re 2 : "If I understand correctly, the main advantage of GD-GNN over NBFNet is that it avoids the cost of GNNs in subgraph extraction, but GD-GNN still needs to perform subgraph extraction for each query. Why is GD-GNN efficient in any scenario? Is it over-claiming?"
>
> We first note that the complexity of geodesic/subgraph extraction is not directly comparable to applying a GNN: geodesic extraction operates on unit integers while GNN operates on $d$-dimensional vectors that are potentially very large, and running a geodesic extraction on a graph takes significantly less amount of time compared to running a GNN on the same graph. In practice, our experiments show that geodesic extraction takes much less time compared to the geodesic pooling process and incurs negligible cost in running time. Therefore, what really matters is how to reduce the number of GNN passes. GD-GNN can thus be understood as a way to use the subgraph information more efficiently which does not require multiple runs of GNN on each extracted subgraph. Instead, it only runs GNN once over the whole graph and does geodesic pooling to inject conditional information.
>
> Also note that the original NBFNet [5] model does not extract subgraphs, instead they run GNN on a labeled full graph. The advantage of GD-GNN over NBFNet is two-fold. First, GD-GNN is a general framework that handles node, link, and graph level tasks, while NBFNet focuses on link prediction. Second, the scenario where NBFNet is efficient is restricted. Only when the query links all start from a single source node can the NBFNet predict multiple links using one GNN pass, whereas the efficiency of GD-GNN is not dependent on the configuration of the query set, since we only run the GNN once and the link information is obtained by pooling the geodesic information.
>
> More formally, here we compare the complexity of GD-GNN, NBFNet, and subgraph-based methods like SEAL[4]. We consider the worst-case scenario where we try to predict $k$ links on a graph with $n$ nodes and $e$ edges. The $k$ links do not share any common nodes. Subgraph GNN methods take $O(kTe)$ for applying $T$-layer GNN onto $k$ links' subgraphs (worst case $O(e)$ edges). NBFNet shares the same complexity $O(kTe)$ because the links do not share common nodes, and the GNN results can not be shared across links as suggested by the authors of NBFNet. Thus, $k$ runs of GNN are necessary for NBFNet. GDGNN takes $O(Te+kn)$ for $O(Te)$ GNN on the full graph and $k$ geodesic pooling, where the number of nodes in a link-level geodesic is bounded by the number of nodes $O(n)$ in the graph. As we have explained in our Re 4 to reviewer CvY7, the practical running time of $O(Te)$ and $O(kn)$ are not on the same level as the GNN is actually quadratic to the embedding dimension $d$ while the geodesic pooling is only linear to $d$. Therefore, in GD-GNN, the more expensive GNN layer complexity $O(Te)$ does not increase linearly w.r.t. query number $k$, which significantly saves the computational time compared to NBFNet or SEAL. This is also demonstrated in Table 3 of our paper.

---

> > ### Author Response · Authors · 2022-08-02
> > **Author Response 2/3**
> >
> > - Re 3 : "The authors are ambiguous about the time complexity of geodesic GNNs. In general, the complexity of geodesic GNNs should between standard GNNs and subgraph extraction. The condition for "GD-GNN maintains the good efficiency of basic GNN models" is that the geodesic readout can be amortized by the cost of the GNN. This is only true for some simple geodesic readout functions, like the two proposed in the paper. The authors should make this more explicit to the readers. Also the time complexity in Appendix B is too optimistic." and "If the pooling layer is a small plain GNN, is geodesic GNNs as slow as SEAL?"
> >
> > Reviewer wYEs is correct and very keen to observe that the complexity of GD-GNN is between that of standard GNN and subgraph GNN. In terms of the pooling layer, reviewer wYEs is also correct that when the geodesic pooling function is more complex, GD-GNN is less efficient. However, Theorem 1 and our experiments have shown that GD-GNN is able to be expressive and effective with a simple readout function. Meanwhile, as the pooling layer gets more complex (e.g. a plain GNN) and overrides the good amortized complexity, it's still upper-bounded by the subgraph methods like SEAL [4] because the subgraphs we consider (geodesics with max cutoff) are usually much smaller than $K$-hop subgraphs. Nevertheless, we agree with reviewer wYEs and we have made this explicit in our revision (Section 3.4).
> >
> >
> > - Re 4 : "Why are higher-order GNNs at least cubic complexity? Could you provide a reference to that?"
> >
> > We have added references [2, 3] to the claim. Note that folklore 2-WL takes $O(n^3)$ and has the same expressiveness as 3-WL, while 3-WL can take up to $O(n^4)$. This is intuitive in that, 3-WL creates $O(n^3)$ node tuples by enumerating all 3-node combinations of the nodes, and each tuple has $O(n)$ neighbors defined by replacing one node in the tuple with another node. Since 3-WL essentially mimics 1-WL but on a larger node and edge set, 1 iteration of 3-WL is $O(n^3*n)=O(n^4)$. 3-GNN uses a subset of the neighbors to reduce running time, but this complexity analysis is still transferable.
> >
> > - Re 5 : "Why is the shortest path viewed as a chain of nodes with embeddings one? What are the embeddings discussed here?"
> >
> > We place this statement there to better motivate the design of horizontal geodesics. Consider a naive GNN that outputs all node embeddings as one, then the horizontal geodesic becomes a vector of one with the length of the shortest path as its dimension. The pooled geodesic representation is exactly the shortest path length. Since naive GNN can be modeled by general GNNs with more complex node embeddings, we say horizontal geodesic extends the shortest path to the neural case.
> >
> > - Re 6 : "You may provide a motivating example here (of when vertical geodesic is better than horizontal one)."
> >
> > We have added an example where vertical geodesic has more expressive power over horizontal geodesic in Appendix H.
> >
> > - Re 7 : "Why are geodesic nodes close to $v$ ignored here?"
> >
> > We thank reviewer for pointing out this implementation choice. On one hand, as shown in Theorem 1, using only geodesic close to the neighbor nodes is sufficient in exceeding 1-WL expressiveness. On the other hand, for efficiency, we can extract all neighbor-side pair-wise geodesic between $v$ and its neighbor nodes using one single-source BFS on $v$, while the geodesic is close to $v$ with respect to other nodes requires more rounds of BFS. This is particularly useful for node-level tasks, where we only need to compute geodesics for part of the nodes.
> >
> > - Re 8 : "To my understanding, this property has been discussed in NBFNet and RED-GNN before. It's better to cite them to provide a fair context."
> >
> > Thanks. We have added the mentioned reference in the revision.
> >
> > - Re 9 : "Appendix B: The time complexity analysis is too optimistic. The authors seem to use the average degree of a node in the full graph / in a local subgraph interchangeably, and assume they are small constants. This may not be true. For example, for a scale-free graph, the average degree may be a constant, but the average degree in a local subgraph may not! This is because every node is likely to be connected to nodes with large degrees, and therefore a lot of subgraphs contain nodes with large degrees. I suggest including the complexity of standard GNNs, SEAL, NBFNet for a fair context."
> >
> > We agree with reviewer wYEs that we should use a more fair setting in the complexity analysis, and we have included some of the analysis in the response above. We will add a dedicated section in the paper to discuss the complexity of different cases under different scenarios.

---

> > > ### Author Response · Authors · 2022-08-02
> > > **Author Response 3/3**
> > >
> > > - Re 10 : "Table 3: Why does NBFNet cause OOM? Could you give an explanation?"
> > >
> > > We use the official implementation of NBFNet. The key reason is that an NBFNet batch contains multiple full graphs, this significantly increases the memory consumption. Moreover, NBFNet uses deeper GNNs and we cannot even fit 1-size batches of the OGBL-PPA dataset into our GPU.
> > >
> > > We will address the compromised efficiency problem when the pooling layer is complex in a limitation section.
> > >
> > > ## Reference
> > >
> > > [1] Li, Pan, et al. "Distance encoding: Design provably more powerful neural networks for graph representation learning." Advances in Neural Information Processing Systems 33 (2020): 4465-4478.
> > >
> > > [2] Morris, Christopher, et al. "Weisfeiler and leman go neural: Higher-order graph neural networks." Proceedings of the AAAI conference on artificial intelligence. Vol. 33. No. 01. 2019.
> > >
> > > [3] Zhang, Muhan, and Pan Li. "Nested graph neural networks." Advances in Neural Information Processing Systems 34 (2021): 15734-15747.
> > >
> > > [4] Zhang, Muhan, and Yixin Chen. "Link prediction based on graph neural networks." Advances in neural information processing systems 31 (2018).
> > >
> > > [5] Zhu, Zhaocheng, et al. "Neural bellman-ford networks: A general graph neural network framework for link prediction." Advances in Neural Information Processing Systems 34 (2021): 29476-29490.

---

> ### Author Response · Authors · 2022-08-07
> **We look forward to your reply**
>
> We thank reviewer wYEs again for the valuable comments to help us improve the paper.
>
> In response to the comments, we address the main concerns as follows:
>
> 1, Node-level experiments. We provide node-level experiments to demonstrate GD-GNN's good performance on node classification.
>
> 2, Complexity analysis. We provide a detailed complexity analysis of GD-GNN and compared it to other methods, which, in turn, shows the efficiency advantage of using GD-GNN.
>
> 3, Complex pooling unit. We explain the complexity of GD-GNN when the pooling unit is complex.
>
> We valued the reviewer's feedback and made a great effort in writing the author's response. Since there are about 3 days left in the discussion phase, would you mind letting us know if our response addresses your concern? If you think there are still other issues, please kindly let us know, we are happy to follow up with you before the discussion phase ends.

---

> > ### Comment · Reviewer_wYEs · 2022-08-08
> > **Response to the authors**
> >
> > Thank the authors for their efforts. As for their response, I am more than satisfied. I update my rating from 6 to 7.
> >
> > I kindly encourage the authors to fulfill their promise if the paper gets accepted. This includes, but not limited to: 1) add node-level experiments 2) clarify that the time complexity relies on relatively simple readout functions 3) credit previous works more explicitly and generously.

---

### Official Review · Reviewer_CvY7 · 2022-07-09

**Rating:** 6
**Confidence:** 4
**Soundness:** 3 good
**Presentation:** 3 good
**Contribution:** 2 fair

**Summary:**

The paper deals with supervised learning on graph structured data for which the most common framework of GNN is limited in expressivity by 1-WL test in terms of its identifying capacity of graphs. More expressive GNN models operate on either higher order tuples of nodes or involve preprocessing of substructures, both of which involve more computation which is not scalable to large graphs. In this paper, the authors propose using the geodesic information between pairs of nodes which includes full shortest-path information. The method includes horizontal geodesic (set of nodes in shortest path) and vertical geodesic (1-hop neighbors which occur in any shortest path). Experimental results on Link Prediction and graph prediction datasets are shown as empirical evidence for the proposed approach.

**Questions:**

Please address the weaknesses mentioned above.


**Limitations:**

I did not find limitations of the approach being discussed in the paper. There is no potential negative societal impact.


**Strengths And Weaknesses:**

### Strengths:
1. The paper addresses an important problem of finding efficient ways of improving the expressivity of the GNNs.
1. **Novelty**: Although the use of shortest path information or distance encodings is shown by some recent works for improving expressivity of GNNs, I find there is good enough novelty in exploring this approach further by incorporating full geodesic information. I find the “vertical” and “horizontal” geodesics proposed to be interesting ways of incorporating the geodesic information explicitly.
1. The geodesic method can distinguish graphs which go beyond distinguishing capacity of mere distance encodings.
1. Experimental results on Link prediction datasets seem to be good.
1. Paper is well written and the presentation is clear. I especially appreciate the intuitive explanations with example graphs to show how “vertical” and “horizontal” geodesics are helpful. However, more annotations in Figure 4 would be helpful.

### Weaknesses:
1. **Permutation invariance**: The method belongs to randomization based techniques to improve expressive power of GNNs since one random shortest path is chosen in computing horizontal geodesic and hence cannot guarantee permutation invariance. This in itself is not a weakness, however, it needs to be acknowledged and the paper needs to be clearly positioned accordingly.
At least some experiments should show comparison with randomization based expressive GNNs like RNI (Abboud  et al. (2020)) and PF-GNN (Dupty et al. (2021))
1. Since the aim of the paper is primarily to go beyond 1-WL GNNs, then it should present experiments where GD-GNN is able to identify graphs not identifiable by 1-WL GNNs like on Circular skip link, Strongly regular graphs etc.. Most papers on the topic of expressive GNNs do exactly this. Such experiments are missing. It would be helpful to see the comparison with higher-order K-GNNs, PPGNN (Maron et al.(2019)), RNI, PF-GNN in order to assess where GD-GNN stands in comparison with other more expressive models.
1. It is not clear how exactly the geodesics are generated efficiently. For ex, how to find vertical geodesics of a pair of nodes. Naively, you would need to  find all shortest paths between nodes and take subset of 1-hop neighbors. Naive way of generating geodesic would take not less than O(n^3) time which is all pairs shortest path. Is  there an efficient way of computing geodesics of all nodes at once?
1. I don’t understand “decoupling” and its usefulness as it is said in line 155-158. What do you mean running GNNs only once? Why can’t we preprocess geodesic information and then use it along with GNN message passing. For ex, preprocess horizontal geodesic, run GNN message passing and pool embeddings from geodesic subset of nodes and repeat this iteratively. Comparing both GNN and geodesic computation, I see that GNN is more efficient than computing geodesics and don’t understand why run GNN only once. Also, do vertical and horizontal geodesics have any learnable parameters?
1. Results on link prediction are impressive. However, the results on graph prediction are not and the baselines used are dated. Since, the motivation of the  paper is to build more expressive GNN, other more expressive models like K-GNN, RNI, PF-GNN should be shown. If the performance is not at par, it is not an issue, however, it should be acknowledged.
1. One of the claim is runtime efficiency compared to other expressive models. However, in worst case, I believe the runtime of GD-GNN is also not less than O(n^3), since the subgraph around a node can be O(n) size. This is same as 3-WL time complexity. Please clarify the worst case runtime in terms of number of nodes n.  Also, in runtime experiments, are you also taking the geodesic computation into account?
1. Ablation study can be expanded. You can test vertical geodesic by instead taking all neighbors of pair of nodes and not just neighbors which are part of shortest paths. This would show if vertical geodesic is important. For horizontal geodesic, you can test does full shortest path is needed or few hop nodes in the shortest path are good enough. You can vary number of hops for ablation.
1. In Section 3.3, the paper first discusses all node, link and graph prediction tasks but doesn’t show experiments on node prediction.
1. For link prediction tasks, I think it would be better to compare with other methods like Position-aware GNNs (You et al.(2019)) which also in a way tries to capture shortest path information.

**Overall**, I find the idea and theme of the paper novel and significant. However, in the present form there are many issues as identified in the weaknesses which makes the paper somewhat in borderline.

### References:
1. Abboud  et al. (2020). "The surprising power of graph neural networks with random node initialization." arXiv preprint arXiv:2010.01179
1. Dupty et al. (2021) "PF-GNN: Differentiable particle filtering based approximation of universal graph representations." International Conference on Learning Representations.
1. You et al.(2019) . "Position-aware graph neural networks." International conference on machine learning. PMLR


########################

Update:
I Thank the authors for the response and additional experiments. Based on the revision, I'm updating the rating and would be happy to see the paper at the conference.

---

> ### Author Response · Authors · 2022-08-02
> **Author Response 1/5**
>
> We thank reviewer CvY7 for their constructive comments on the paper. We address their concerns and questions as follows.
>
> - Re 1 : "Permutation invariance: The method belongs to randomization-based techniques to improve the expressive power of GNNs since one random shortest path is chosen in computing horizontal geodesic and hence cannot guarantee permutation invariance. This in itself is not a weakness, however, it needs to be acknowledged and the paper needs to be clearly positioned accordingly. At least some experiments should show comparison with randomization-based expressive GNNs like RNI (Abboud et al. (2020)) and PF-GNN (Dupty et al. (2021))"
>
> We agree that horizontal geodesic also has randomness, however, horizontal geodesic and randomness-based GNNs have very different motivations behind their randomness. The motivation behind RNI [1] and PF-GNN [2] is to use randomness (in particular, uniqueness of node features/coloring) to achieve universal approximation of graph functions. In contrast, we use horizontal geodesic to model the conditional relationship between two nodes and learn their relative information. We randomly choose one horizontal geodesic only for computational efficiency. In other words, our randomness is not for the purpose of enhancing expressiveness (at the cost of breaking permutation invariance), but simply for reducing computational complexity. Nevertheless, we add a comparison with RNI and PF-GNN in Re 3.
>
> - Re 2 : "However, more annotations in Figure 4 would be helpful."
>
> We appreciate reviewer CvY7's feedback and we've included edge coloring for better readability in the revision.
>
> - Re 3 : "Since the aim of the paper is primarily to go beyond 1-WL GNNs, then it should present experiments where GD-GNN is able to identify graphs not identifiable by 1-WL GNNs like on Circular skip link, Strongly regular graphs etc.. Most papers on the topic of expressive GNNs do exactly this. Such experiments are missing. It would be helpful to see the comparison with higher-order K-GNNs, PPGNN (Maron et al.(2019)), RNI, PF-GNN in order to assess where GD-GNN stands in comparison with other more expressive models."
>
> We thank reviewer CvY7 for suggesting experiments on synthetic datasets to verify the expressiveness of our GNN. We present the results on two synthetic datasets as follows:
>
> |             |  EXP  |  CSL  |
> |-------------|:-----:|:-----:|
> | GIN         |  50.0 |  10.0 |
> | RNI         |  99.7 |   16.0 |
> | PF-GNN       | 100.0 | 100.0 |
> | 2-GDGNN-Ver |  50.0 |  30.0 |
> | 3-GDGNN-Ver | 100.0 |  60.0 |
> | 4-GDGNN-Ver | 100.0 | 100.0 |
>
> N-GD-GNN means that the cutoff distance and the number of layers are N. As we can see from the experiment, with a sufficient number of layers, GDGNN achieves comparable expressiveness to other more-expressive GNNs, we have also included this result in the revision (Section 5.2).
>
> - Re 4 : "It is not clear how exactly the geodesics are generated efficiently. For ex, how to find vertical geodesics of a pair of nodes. Naively, you would need to find all shortest paths between nodes and take subset of 1-hop neighbors. Naive way of generating geodesic would take not less than O(n^3) time which is all pairs shortest path. Is there an efficient way of computing geodesics of all nodes at once?"
>
> We use BFS and neighbor set intersections to extract geodesics. Since we set $k$ as the cutoff distance, the complexity of geodesic extraction is usually much smaller than $O(n^3)$. Meanwhile, we can always precompute and cache the geodesics which allows reusing and fast geodesic pooling during inference. In practice, our implementation extracts geodesics for nodes on the fly. Nevertheless, our experiments show that geodesic extraction takes much less time compared to the geodesic pooling process and incurs negligible running time. On our machine, extracting geodesics of batch 32 takes ~0.0004 sec, while geodesic pooling takes ~0.0024 sec on the OGBG-MOLHIV dataset.

---

> > ### Author Response · Authors · 2022-08-02
> > **Author Response 2/5**
> >
> > - Re 5 : "One of the claim is runtime efficiency compared to other expressive models. However, in worst case, I believe the runtime of GD-GNN is also not less than O(n^3), since the subgraph around a node can be O(n) size. This is same as 3-WL time complexity. Please clarify the worst case runtime in terms of number of nodes n. Also, in runtime experiments, are you also taking the geodesic computation into account?"
> >
> > Note that folklore 2-WL takes $O(n^3)$ and has the same expressiveness as 3-WL, while 3-GNN adapts 3-WL and can take up to $O(n^4)$. This is intuitive in that, 3-WL creates $O(n^3)$ node tuples by enumerating all 3-node combinations of the nodes, each tuple has $O(n)$ neighbors defined by replacing one node in the tuple with another node. Since 3-WL essentially mimics 1-WL but on a larger node and edge set, 1 iteration of 3-WL is $O(n^3*n)=O(n^4)$. 3-GNN uses a subset of the neighbors to reduce running time, but this complexity analysis is still transferable.
> >
> > On the other hand, we note that the complexity of geodesic extraction and GNN cannot be directly compared as their computation units are different: geodesic extraction operates on unit integers while GNN operates on $d$-dimensional vectors that are potentially very large, and running a geodesic extraction on a graph takes significantly less amount of time compared to running a GNN on the same graph. Therefore, we compare the worst-case complexity of the GNN process. Consider using $T$ layers of message passing. For subgraph-based GNNs, to infer on a graph with $n$ nodes and $e$ edges, the complexity is $O(nTe)$ for applying $O(Te)$ complexity 1-WL GNN onto $n$ subgraphs (each subgraph is a full graph with $e$ edges in the worst case). For GD-GNN, the complexity is $O(Te+ne)=O(ne)$, due to applying the $T$-layer 1-WL GNN once to the whole graph, and pooling $O(n)$ node-level vertical geodesics. Note that the number of nodes in a node's vertical geodesic is bounded by the number of edges $e$ in the graph. The comparison shows that GD-GNN helps amortize the expensive computation cost, $T$ versus $1$, of applying a GNN. If we further consider GNN's layer complexity w.r.t. the embedding dimension $d$ (mostly quadratic due to linear transformation/MLP over node embeddings, e.g. GCN and GIN), we will have $O(nTed^2)$ for subgraph GNNs, but $O(Ted^2+ned)$ for GD-GNN (the geodesic pooling is linear to $d$). Thus, GD-GNN can yield even more efficiency advantages for large $d$.
> >
> > We also emphasize the significant efficiency advantage of GD-GNN on link prediction tasks. To infer $k$ links, subgraph GNN methods take $O(kTe)$ for applying $O(Te)$ GNN onto $k$ links' subgraphs. GDGNN takes $O(Te+kn)$ for one $O(Te)$ GNN on the full graph and $k$ geodesic pooling. Unlike in graph/node level tasks, link level geodesic is bounded by the number of nodes in the graph. As we can see, in GD-GNN, the more expensive GNN layer complexity does not increase linearly w.r.t. query number $k$, which significantly saves computational time. A more detailed complexity analysis can be found in the revision (Appendix B).
> >
> > The runtime results in the paper include geodesic extraction time.

---

> > > ### Author Response · Authors · 2022-08-02
> > > **Author Response 3/5**
> > >
> > > - Re 6 : "I don't understand "decoupling" and its usefulness as it is said in line 155-158. What do you mean running GNNs only once? Why can't we preprocess geodesic information and then use it along with GNN message passing. For ex, preprocess horizontal geodesic, run GNN message passing and pool embeddings from geodesic subset of nodes and repeat this iteratively. Comparing both GNN and geodesic computation, I see that GNN is more efficient than computing geodesics and don't understand why run GNN only once. Also, do vertical and horizontal geodesics have any learnable parameters?"
> > >
> > > The reviewer suggests an interesting direction where we apply geodesic pooling after each layer of GNN and use the pooled node embedding as the input to the next GNN layer. This can potentially increase the power of GDGNN, yet slightly violates our motivation to inject conditional information to any node-level GNN in a post-processing way without modifying its message passing process. Nevertheless, we will take this as a promising future direction for GDGNN. On the other hand, the 'decoupling' is relative to other subgraph-based GNNs, and not to plain GNN itself. In subgraph-based GNN methods, the T-layer GNN is coupled with the extracted subgraphs, and for each subgraph, we run the T-layer GNN once, while decoupled GDGNN can first apply an arbitrary GNN once to the full graph and use geodesics of different learning targets to inject conditional information without running multiple times of GNN like in subgraph-based methods. Hence the geodesic extraction process and the GNN process are decoupled. As we've discussed in Re 4, applying GNN multiple times on subgraphs does increase the complexity from $O(ne)$ to $O(nTe)$, and GD-GNN is more efficient than the $O(n^4)$ 3-GNN. That's why we choose to only run GNN once on the full graph and rely on the good theoretical property of geodesics to efficiently generate target representations from the computed embeddings of the GNN. Vertical and horizontal geodesics do not have learnable parameters as they are essentially a selection rule to inject higher expressiveness.
> > >
> > > - Re 7 : "Results on link prediction are impressive. However, the results on graph prediction are not and the baselines used are dated. Since, the motivation of the paper is to build more expressive GNN, other more expressive models like K-GNN, RNI, PF-GNN should be shown. If the performance is not at par, it is not an issue, however, it should be acknowledged."
> > >
> > > Since our primary motivation is to circumvent the computation burden of subgraph-based GNNs, so we originally chose representative models GNN-AK and NGNN for comparison. We appreciate the reviewer for pointing us to other more expressive models. And apart from the comparison of the synthetic dataset, we also include the comparison of real-world datasets.
> > >
> > > |            |   OGBG-MOLHIV (test)  |
> > > |------------|:-----------------:|
> > > | GIN     |   75.58±1.40   |
> > > | RNI     |   75.46±2.27   |
> > > | GSN       |   77.99±1.00   |
> > > | GIN-AK | 78.22±0.75 |
> > > | NGNN | 78.34±1.86 |
> > > | Directional GSN | **80.39±0.40** |
> > > | PF-GNN | 80.15±0.68 |
> > > | GD-GNN-Ver | 78.43±1.02 |
> > >
> > >
> > > As we can see, GD-GNN's performance is lower than PF-GNN and Directional GSN (so are GIN-AK and NGNN). This might indicate that pairwise conditional information is less effective for graph tasks than for link tasks. Nevertheless, GD-GNN still outperforms the basic GIN a lot, and as GD-GNN is a general framework that can be applied to any basic GNN, the performance might be able to be further improved.

---

> > > > ### Author Response · Authors · 2022-08-02
> > > > **Author Response 4/5**
> > > >
> > > > - Re 8 : "Ablation study can be expanded. You can test vertical geodesic by instead taking all neighbors of pair of nodes and not just neighbors which are part of shortest paths. This would show if vertical geodesic is important. For horizontal geodesic, you can test does full shortest path is needed or few hop nodes in the shortest path are good enough. You can vary number of hops for ablation."
> > > >
> > > > For vertical geodesics, the difference between our base GIN and GD-GNN shows that vertical GD indeed improves the expressiveness beyond 1-WL. The variant where all neighbors are used is essentially adding another plain GNN layer. Nevertheless, we show the comparison as follows.
> > > >
> > > > |         | MUTAG   | PROTEINS    | OGBG-MOLHIV    |
> > > > |---------|:-------:|:-----------:|:--------------:|
> > > > | GD-GNN-Vert-FullNei |  88.1  |    71.9    |      76.4     |
> > > > | GD-GNN-Vert-Deg |  89.4  |    73.6    |      78.4     |
> > > >
> > > > FullNei represents the variant where all nodes around the neighbors are used to generate geodesics. We can see that vertical geodesic still outperforms FullNei. This shows that FullNei weighs in as an extra layer of GNN which can be covered by hyperparameter search.
> > > >
> > > > For horizontal geodesics, we present results where part of the horizontal geodesic is used as the reviewer suggests.
> > > >
> > > > |         | Cora | OGBL-COLLAB |
> > > > |---------|:-------:|:-----------:|
> > > > | 4-GDGNN-Hor |  94.40  |    52.13    |
> > > > | 4-GDGNN-Partial |  92.73  |    52.06    |
> > > > | 5-GDGNN-Hor |  94.37  |    52.18    |
> > > > | 5-GDGNN-Partial |  92.68  |    51.42    |
> > > >
> > > > Partial represents geodesics where only the head/tail and head/tail's direct neighbors on the shortest path are used to form the geodesic. N-GDGNN means the cutoff distance is $N$. We can see that without the full horizontal geodesic, the performance on the Cora dataset indeed dropped by ~1.5%. While the difference is not as significant, we still see that horizontal geodesics outperforms Partial on the OGBL-COLLAB dataset. We have included the results in the revised ablation study.
> > > >
> > > > - Re 9 : "In Section 3.3, the paper first discusses all node, link and graph prediction tasks but doesn't show experiments on node prediction."
> > > >
> > > > We didn't include node-level tasks initially because graph-level tasks can also reflect a method's ability to learn structures around a node, but we agree with the reviewer and include the node-level task results as follows.
> > > >
> > > > |            |   Bra.-Airports   |   Eur.-Airports   |    USA-Airports   |
> > > > |------------|:-----------------:|:-----------------:|:-----------------:|
> > > > | DE-GNN     |   75.37±3.25   | **58.41±3.20** |   64.16±1.70   |
> > > > | GIN       |   72.83±3.57   |   53.84±3.94   |   57.43±2.13   |
> > > > | GDGNN-ver | **78.61±2.20** |   53.97±3.50   | **64.36±1.62** |
> > > >
> > > > We compare to DE-GNN [5] on the airport datasets, as DE-GNN also uses distance information. The task is to predict passenger flow levels solely from the flight traffic network. We choose these datasets because the node classes reflect their structural roles in the network instead of community identifiers as in commonly used citation networks (such as Cora and Citeseer). Only in these datasets, graph structure matters, and more expressive GNNs are useful [5]. DE-GNN has several variants and we take their best result here. From the experiment, we can see that GD-GNN achieved very competitive results compared to distance-encoded GNN, and significantly improved the performance of base GIN on two datasets, which experimentally verify our claim that node-level geodesic is more expressive than plain GNNs. The results are included in the revision (Appendix E).

---

> > > > > ### Author Response · Authors · 2022-08-02
> > > > > **Author Response 5/5**
> > > > >
> > > > > - Re 10 : "For link prediction tasks, I think it would be better to compare with other methods like Position-aware GNNs (You et al.(2019)) which also in a way tries to capture shortest path information."
> > > > >
> > > > > We agree with Reviewer CvY7, and present link prediction results compared to other distance-related GNN.
> > > > >
> > > > > |           |    C.elegans    |        NS       |        PB       |
> > > > > |-----------|:---------------:|:---------------:|:---------------:|
> > > > > | PGNN      |   78.20±0.33   |   82.76±0.59   |   90.47±0.60   |
> > > > > | SEAL      |   88.26±0.56   |   98.55±0.32   |   94.18±0.57   |
> > > > > | DE-GNN    | **90.05±0.26** | **99.43±0.63** |   94.95±0.37   |
> > > > > | GDGNN-hor |   89.83±0.70   |   98.65±0.48   | **96.14±0.73** |
> > > > > | GDGNN-ver |   87.88±0.42   |   98.10±0.26   |   94.43±0.39   |
> > > > >
> > > > > We follow DE-GNN's experiment setting. Likewise, DE has several variants and we take their best results. From the experiment, we can see that GD-GNN achieves very competitive results compared to distance-encoding GNN and SEAL [7], and outperforms PGNN [8] significantly on all datasets. We suspect the key reason is that PGNN relies on relative positions to the anchors, but the anchors that PGNN randomly chooses are not necessarily representative for all links. In contrast, the geodesic information is directly associated with each link. The results are included in the revision (Appendix D).
> > > > >
> > > > >
> > > > >
> > > > > ## Reference
> > > > >
> > > > >
> > > > > [1] Abboud, Ralph, et al. "The surprising power of graph neural networks with random node initialization." arXiv preprint arXiv:2010.01179 (2020).
> > > > >
> > > > > [2] Dupty, Mohammed Haroon, Yanfei Dong, and Wee Sun Lee. "PF-GNN: Differentiable particle filtering based approximation of universal graph representations." International Conference on Learning Representations. 2021.
> > > > >
> > > > > [3] Zhang, Muhan, et al. "Labeling trick: A theory of using graph neural networks for multi-node representation learning." Advances in Neural Information Processing Systems 34 (2021): 9061-9073.
> > > > >
> > > > > [4] You, Jiaxuan, et al. "Identity-aware graph neural networks." Proceedings of the AAAI Conference on Artificial Intelligence. Vol. 35. No. 12. 2021.
> > > > >
> > > > > [5] Zhang, Muhan, and Pan Li. "Nested graph neural networks." Advances in Neural Information Processing Systems 34 (2021): 15734-15747.
> > > > >
> > > > > [6] Li, Pan, et al. "Distance encoding: Design provably more powerful neural networks for graph representation learning." Advances in Neural Information Processing Systems 33 (2020): 4465-4478.
> > > > >
> > > > > [7] Zhang, Muhan, and Yixin Chen. "Link prediction based on graph neural networks." Advances in neural information processing systems 31 (2018).
> > > > >
> > > > > [8] You, Jiaxuan, Rex Ying, and Jure Leskovec. "Position-aware graph neural networks." International conference on machine learning. PMLR, 2019.

---

> ### Author Response · Authors · 2022-08-07
> **We look forward to your reply**
>
> We thank reviewer CvY7 again for the inspiring comments to help us improve the paper.
>
> In response to the comments, we address the main concerns as follows:
>
> 1, Verification of theoretical results. We present the results of GDGNN on synthetic datasets to verify >1-WL expressiveness. We also present a comparison to more expressive GNN on real-world datasets.
>
> 2, Randomness in the system. We clarify the difference in motivation between horizontal GD-GNN and other random GNNs.
>
> 3, Complexity. We provide a detailed complexity analysis of GD-GNN in the worst case for node/link/graph level tasks.
>
> 4, Ablation study. We expand the ablation to evaluate the effectiveness of different components of GD-GNN more thoroughly.
>
> 5, Link prediction comparison. We provide additional experiments comparing GD-GNN to other distance-based GNN.
>
> We valued the reviewer's feedback and made a great effort in writing the author's response. Since there are about 3 days left in the discussion phase, would you mind letting us know if our response addresses your concern? If you think there are still other issues, please kindly let us know, we are happy to follow up with you before the discussion phase ends.

---

> ### Comment · Reviewer_CvY7 · 2022-08-07
> **Reviewer acknowledgement**
>
> I thank the authors for the detailed response and additional experiments. Most of my concerns on experiments are cleared. Hence, I have updated my rating and am now inclined for acceptance of the paper.

---

### Official Review · Reviewer_SHs4 · 2022-07-12

**Rating:** 7
**Confidence:** 4
**Soundness:** 3 good
**Presentation:** 3 good
**Contribution:** 3 good

**Summary:**

This paper proposes a novel mechanism to efficiently improve learned graph representations. It proposes an independent second layer built upon base GNN representations that can improve the performance and expressivity of the base GNN. This mechanism is the Geodesic pooling layer. The horizontal version works by pooling the node representations along the shortest path between a pair of nodes. The vertical version works by pooling the nearest neighbors to the target node along any shortest path between the nodes of interest.

The paper then proposes different techniques for using these geodesics for node-level, edge-level, and graph-level tasks.

For experimentation, they do experimental testing across different domains. They test on link prediction tasks for knowledge graphs and link prediction on OGB datasets. They also test on graph classification using TU and OGB datasets. Node classification tasks are not experimentally tested.

The results show very good results on edge level and graph level tasks.

There is also a bit of theoretical analysis showing that the model can distinguish certain types of graphs (r-regular) with a high and controllable likelihood. In addition there are good synthetic examples where existing works would fail to distinguish certain graphs or edges.

**Questions:**

1) For node level tasks, how is the geodesic approach any different to sampling neighbors of a node and running a GNN? (maybe with included feature indicating node of interest). Does not seem like this approach would confer any advantage or difference in this realm.

2) Since you use a max distance cutoff, how often do the graphs have links that are beyond the max distance? Does the model still predict these links in practice or does the model learn to never predict links between "disconnected" nodes?

3) How would the presence of node features affect or limit the analysis?

**Limitations:**

Limitations/questions surround the node level tasks, analysis under the presence of node features.

**Strengths And Weaknesses:**

Originality:

++ Work is original. Ideas are novel.

-- Should be comparison to how this differs from path based reasoning methods (many exist for KG completion)

Quality:

++ The work is a complete work and is of generally high quality. The ideas are well motivated.

++ This approach seems very well attuned to link level tasks and would certainly be of interest to people working with these tasks (e.g. Knowledge Graph Completion). It also seems like it is better than the baselines in graph completion.

-- Does not seem like there would be any advantage to using this on node-level tasks as there is basically no difference to existing GNNs. It is essentially sampling neighbors of a node and then applying another GNN on top of that.

-- Also, no testing of node level tasks

-- The KG completion task should not necessarily be considered state of the art. I think the baselines are fair for comparison, but there are other methods that take advantage of the textual data available via language modeling for better inductive knowledge graph completion. I think the authors should mention that they exclude text based inductive models for KG completion. Or they could add these baselines and see how they compare?

Including:

Wang, Bo et al. “Structure-Augmented Text Representation Learning for Efficient Knowledge Graph Completion.” Proceedings of the Web Conference 2021 (2021): n. pag.

Daza, Daniel et al. “Inductive Entity Representations from Text via Link Prediction.” Proceedings of the Web Conference 2021 (2021): n. pag.

-- The theoretical analysis is limited to to r-regular graphs with no node features. This is not a common occurance, but does slightly support the idea that there may be an advantage over standard GNNs. However, in general, this work does not deal well with how node features affect the analysis. Authors state "For simplicity, we use homogeneous graphs to illustrate our ideas and methods" but the analysis likely does not hold up as well for non homogenous graphs.

-- Small note: GNNs often use only a few layers because there is no performance advantage (e.g. over smoothing), not because of computational cost. (line 174)

+- Small note: The horizontal geodesic seems like a good idea for connecting far nodes, however, there is still a max depth.

Clarity:

++ The writing is generally precise and clear.

-- The use of the v1 and v3 splits  for inductive kg completion is slightly cherry picked (the full table is in the appendix)

Grammar/Spelling:

Figure 4: distingshui -> distinguish

---

> ### Author Response · Authors · 2022-08-02
> **Author Response 1/3**
>
> - Re 1 : "Does not seem like there would be any advantage to using this on node-level tasks as there is basically no difference to existing GNNs. It is essentially sampling neighbors of a node and then applying another GNN on top of that." and "For node level tasks, how is the geodesic approach any different to sampling neighbors of a node and running a GNN? (maybe with included feature indicating node of interest). Does not seem like this approach would confer any advantage or difference in this realm."
>
> We want to first clarify that for node tasks, we only run the GNN once on the original graph. The node representation generation is through the geodesic pooling layer, and no other GNN is applied to the representations generated. Specifically, after obtaining the node embedding from the GNN, for each query node $v$, we take its k-hop neighbor set $S$, and for each neighbor $s\in S$, we extract a pair-wise geodesic between $v$ and $s$. Then $|S|$ pair-wise geodesics can be pooled by an injective pooling function to form the node representation.
>
> Theoretically, GD-GNN utilized geodesic information and can distinguish between nodes that 1-WL can not distinguish. Specifically, 1-WL GNN can not distinguish between two different nodes in r-regular graphs while GD-GNN can, which is shown in Theorem 1. For node $v$ and its neighbor $s$, the vertical geodesic $P_s$ of $s$ is formed by the nodes in $\mathcal{N}_s$ that are also on the shortest paths between $s$ and $v$, note that $|P_s|\leq|\mathcal{N}_s|$. Consider a feature-less r-regular graph, $|\mathcal{N}_u|=r$ for all $u\in V$, for two non-isomorphic nodes $v$ and $v'$, GNN still assigns them the same representations, whereas the neighbor nodes $S$ of $v$ have a different set of $|P_s|$ from the $|P_s'|$ of neighbor set $S'$ for $v'$, and vertical Geodesic is able to reflect on the difference to tell the two nodes apart. More details can be found in the proof of Theorem 1 in Appendix A.
>
> On the other hand, compared to more-expressive subgraph GNNs, GD-GNN only needs to run the GNN once and is more efficient. Reviewer suggests sampling neighbors of a node and running GNN on top of them. This can be seen as a randomized version of NGNN [3], which is a subgraph-based method. Compared to subgraph-based GNNs like ID-GNN [4] and NGNN, GD-GNN only applies the GNN once, which brings good efficiency. Specifically, consider the problem where we need to infer $k$ nodes in a graph with $n$ nodes and $e$ edges. In worst-case, the complexity for a T-layer subgraph-based GNN is $O(kTe)$, because it applies a $T$-layer GNN on $k$ nodes' subgraphs, where each subgraph contains $e$ edges in the worst case (the full graph). GD-GNN's worst-case complexity is $O(Te+ke)$, $O(Te)$ is for applying the GNN once to the full graph, and the number of nodes in a vertical geodesic is bounded by the number of edges hence the pooling takes $O(ke)$. Hence, GD-GNN is more efficient than subgraph GNNs. If we consider GNN's layer complexity w.r.t. the embedding dimension $d$ (mostly quadratic due to linear transformation/MLP over node embeddings, e.g. GCN and GIN), we will have $O(kTed^2)$ for subgraph GNNs, but $O(Ted^2+ked)$ for GD-GNN (the geodesic pooling is linear to $d$). Thus, the efficiency advantage of GD-GNN is more significant as shown in our experiments. A more detailed complexity analysis can be found in the revision (Appendix B).
>
> Meanwhile, we agree with reviewer SHs4 that we should include the results of node-level tasks to verify our claims.
>
> |            |   Bra.-Airports   |   Eur.-Airports   |    USA-Airports   |
> |------------|:-----------------:|:-----------------:|:-----------------:|
> | DE-GNN     |   75.37±3.25   | **58.41±3.20** |   64.16±1.70   |
> | GIN       |   72.83±3.57   |   53.84±3.94   |   57.43±2.13   |
> | GDGNN-ver | **78.61±2.20** |   53.97±3.50   | **64.36±1.62** |
>
> We compare to DE-GNN [5] on the airport datasets, as DE-GNN also uses distance information. The task is to predict passenger flow levels solely from the flight traffic network. We choose these datasets because the node classes reflect their structural roles in the network instead of community identifiers as in commonly used citation networks (such as Cora and Citeseer). Only in these datasets, graph structure matters, and more expressive GNNs are useful [5]. DE-GNN has several variants and we take their best result here. From the experiment, we can see that GD-GNN achieved very competitive results compared to distance-encoded GNN, and significantly improved the performance of base GIN on two datasets, which experimentally verify our claim that node-level geodesic is more expressive than plain GNNs. The results are included in the revision (Appendix E).

---

> > ### Author Response · Authors · 2022-08-02
> > **Author Response 2/3**
> >
> > - Re 2 : "Should be comparison to how this differs from path based reasoning methods (many exist for KG completion)."
> >
> > Note that vertical geodesic does not explicitly extract paths and hence is different from path-based KG completion models, but we agree that horizontal geodesic is similar to other path-based methods and should be more carefully compared. [1] uses CNN and BiLSTM to summarize the path information, however, their model requires fixed node embedding like other transductive KG completion models, while horizontal geodesic is based on graph structure generated from GNN and hence is inductive. [2] learns relational path information by enumerating all paths between links and training a set of embeddings for each different type of relational path, while horizontal GD-GNN uses GNN to capture the graph structure and pool node embeddings along the geodesic without training relational-path-specific embeddings. We have added this discussion into Section 4 of the updated paper (in blue).
> >
> > - Re 3 : "The KG completion task should not necessarily be considered state of the art. I think the baselines are fair for comparison, but there are other methods that take advantage of the textual data available via language modeling for better inductive knowledge graph completion. I think the authors should mention that they exclude text-based inductive models for KG completion. Or they could add these baselines and see how they compare?"
> >
> > Reviewer SHs4 is correct. We follow [6], and focus on comparing with models solely using the graph, and we have stated this more explicitly in our revision (see Section 5). We believe models that employ textual information of the knowledge graph are beyond the context of graph structural learning.
> >
> > - Re 4 : "The theoretical analysis is limited to r-regular graphs with no node features. This is not a common occurance, but does slightly support the idea that there may be an advantage over standard GNNs. However, in general, this work does not deal well with how node features affect the analysis. Authors state "For simplicity, we use homogeneous graphs to illustrate our ideas and methods" but the analysis likely does not hold up as well for non homogenous graphs." and "How would the presence of node features affect or limit the analysis?"
> >
> > Yes, we focus our analysis on graphs with no node features. This is because using node features only makes the model more expressive and distinguishes more nodes/links/graphs. An analysis of non-attributed graphs provides a lower bound for the expressive power and is standard in the literature (e.g., NGNN, DE, etc.). And our results show that purely based on graph structures, we can already surpass 1-WL by distinguishing most regular graphs. As each update is an injective mapping of both structures and features, adding node features will at least not decrease the model expressiveness and our conclusion of >1-WL expressiveness still holds.
> >
> > - Re 5 : "Small note: GNNs often use only a few layers because there is no performance advantage (e.g. over smoothing), not because of computational cost."
> >
> > We agree with reviewer SHs4 that many GNNs suffer from the over-smoothing effect when the model is deep. However, as recent work [7] points out, a deep GNN is actually helpful in learning more accurate node embeddings with some decoupling techniques. Also as pointed out by [8], over smoothing is not an issue if we use learnable graph filters (instead of the low-pass filter as in GCN). In such cases, a deep model is still much desired and a method that does not need to run the large deep GNN multiple times (GDGNN) is more computationally friendly.
> >
> > - Re 6 : "The use of the v1 and v3 splits for inductive kg completion is slightly cherry picked (the full table is in the appendix)."
> >
> > We will include the full table in the camera-ready version with more spaces. On WN18RR inductive dataset, we still outperform other methods. On FB15K237 V2 and V4 inductive dataset, where GDGNN is out-performed by NBFNet, we have \<1% relative performance decrease.

---

> > > ### Author Response · Authors · 2022-08-02
> > > **Author Response 3/3**
> > >
> > > - Re 7 : "Since you use a max distance cutoff, how often do the graphs have links that are beyond the max distance? Does the model still predict these links in practice or does the model learn to never predict links between "disconnected" nodes?"
> > >
> > > Reivewer SHs4 is correct that different max cutoff distance indeed makes a difference. We present the results of using different max-cutoff here.
> > >
> > > |         | Cora(L) | OGBL-PPA(L) | OGBG-MOLHIV(G) |
> > > |---------|:-------:|:-----------:|:--------------:|
> > > | 1-GDGNN |  82.46  |    21.79    |      75.68     |
> > > | 2-GDGNN |  91.80  |    43.57    |      78.13     |
> > > | 3-GDGNN |  93.22  |    45.61    |      78.42     |
> > > | 4-GDGNN |  94.93  |    44.98    |      78.36     |
> > >
> > > (L) Represents the link prediction task, and (G) represents the graph classification task. N-GDGNN means GNN with different max-cutoff distances, we use vertical GD-GNN with 3-layers of base GNN. For the link prediction task (Cora in particular), as the max-cutoff distance increases, the performance increases, this is because more links can be connected by geodesic extraction, and distance itself is already a good indicator of the likelihood of the link. However, disconnected nodes are not always predicted negative. When disconnected, the link will have zero geodesic representation, but **still have meaningful node representations from the base GNN**. In such cases, GD-GNN degenerates embeddings similar to models like VGAE [9], which is still able to statistically learn the probability of a link based on the node structure around the two nodes of the link. The choice of max-distance is really data-dependent, as we can see in the results of OGBL-PPA and OGBG-MOLHIV, 4-GDGNN is worse than 3-GDGNN, and the actual number can be determined by hyperparameter tuning. In general, cutoff distance resembles the max number of hops in the subgraph extraction process.
> > >
> > > Thanks again for all the comments to make our paper better. We will correct grammar and spelling errors in our revision.
> > >
> > > ## Reference
> > >
> > > [1] Jagvaral, Batselem, et al. "Path-based reasoning approach for knowledge graph completion using CNN-BiLSTM with attention mechanism." Expert Systems with Applications 142 (2020): 112960.
> > >
> > > [2] Wang, Hongwei, Hongyu Ren, and Jure Leskovec. "Relational message passing for knowledge graph completion." Proceedings of the 27th ACM SIGKDD Conference on Knowledge Discovery & Data Mining. 2021.
> > >
> > > [3] Zhang, Muhan, and Pan Li. "Nested graph neural networks." Advances in Neural Information Processing Systems 34 (2021): 15734-15747.
> > >
> > > [4] You, Jiaxuan, et al. "Identity-aware graph neural networks." Proceedings of the AAAI Conference on Artificial Intelligence. Vol. 35. No. 12. 2021.
> > >
> > > [5] Li, Pan, et al. "Distance encoding: Design provably more powerful neural networks for graph representation learning." Advances in Neural Information Processing Systems 33 (2020): 4465-4478.
> > >
> > > [6] Zhu, Zhaocheng, et al. "Neural bellman-ford networks: A general graph neural network framework for link prediction." Advances in Neural Information Processing Systems 34 (2021): 29476-29490.
> > >
> > > [7] Zeng, Hanqing, et al. "Decoupling the depth and scope of graph neural networks." Advances in Neural Information Processing Systems 34 (2021): 19665-19679.
> > >
> > > [8] Chien, Eli, et al. "Adaptive Universal Generalized PageRank Graph Neural Network." International Conference on Learning Representations.
> > >
> > > [9] Kipf, Thomas N., and Max Welling. "Variational graph auto-encoders." arXiv preprint arXiv:1611.07308 (2016).

---

> ### Author Response · Authors · 2022-08-07
> **We look forward to your reply**
>
> We thank reviewer SHs4 again for the constructive comments to help us improve the paper.
>
> In response to the comments, we address the main concerns as follows:
>
> 1, Node-level task experiment. We provide node-level geodesic experiments on air traffic datasets.
>
> 2, Node-level advantage. We clarify how GD-GNN is used in node-level tasks and provide a detailed complexity analysis of the node-level task to show the better efficiency of GD-GNN.
>
> 3, Comparison to other knowledge graph completion methods. We clarify why we believe we should compare GD-GNN to other KG completion methods without using textual information. We also include a more comprehensive comparison to other path-based KG completion methods in Section 4.
>
> We valued the reviewer's feedback and made a great effort in writing the author's response. Since there are about 3 days left in the discussion phase, would you mind letting us know if our response addresses your concern? If you think there are still other issues, please kindly let us know, we are happy to follow up with you before the discussion phase ends.

---

### Meta-Review · Area_Chair_BJBJ · 2022-08-31

**Recommendation:** Accept
**Confidence:** Less certain

**Metareview:**

I must say that I am impressed with authors' diligence in the rebuttal, adding new experiments and revising the paper. The paper had many issues in the original submission, although it was still well-written. The reviewers are more convinced about the paper after the rebuttal. I would recommend a (weak) acceptance with some (unaddressed) concerns.

To improve the expressivity power of GNN, in this paper the authors developed a mechanism (Geodesic pooling) for improving learnt graph representations, by using an independent second layer based on GNN.  They presented good results on edge level and graph level tasks. The reviewers believe the method is novel and worth the acceptance if the many issues raised could be resolved.

More details: GNN are prone to map different nodes in a graph to the same embedding vectors in the embedding space even if they are far away from each other in the graph. Specifically, if the local structure of two nodes are similar, the convolution layer maps both of them to the same embedding vector. The paper proposes to include geodesic information to help to distinguish nodes with similar local structures. However, for a graph classification task, it requires computing the shortest path between two many pairs of nodes which could be computationally expensive. In addition, the paper does not cite and compare to other important baseline. For instance, a simple and effective solution is to simply provide geometrical embedding vectors as initial node attributes to the GNN. Some works that can be cited and compared against:

Graph Neural Networks with Learnable Structural and Positional Representations

The Necessity of Geometrical Representation for Deep Graph Analysis

Node Proximity Is All You Need: Unified Structural and Positional Node and Graph Embedding

Inductive Graph Embeddings through Locality Encodings


**Award:**

No

---

### Decision · Program_Chairs · 2022-09-14

Accept